# Feed-forward regulation adaptively evolves via dynamics rather than topology when there is intrinsic noise

Kun Xiong [1], Alex K. Lancaster [2], Mark L. Siegal [3] & Joanna Masel [4]

In transcriptional regulatory networks (TRNs), a canonical 3-node feed-forward loop (FFL) is hypothesized to evolve to filter out short spurious signals. We test this adaptive hypothesis against a novel null evolutionary model. Our mutational model captures the intrinsically high prevalence of weak affinity transcription factor binding sites. We also capture stochasticity and delays in gene expression that distort external signals and intrinsically generate noise. Functional FFLs evolve readily under selection for the hypothesized function but not in negative controls. Interestingly, a 4-node "diamond" motif also emerges as a short spurious signal filter. The diamond uses expression dynamics rather than path length to provide fast and slow pathways. When there is no idealized external spurious signal to filter out, but only internally generated noise, only the diamond and not the FFL evolves. While our results support the adaptive hypothesis, we also show that non-adaptive factors, including the intrinsic expression dynamics, matter.

[1] Department of Molecular and Cellular Biology, University of Arizona, Tucson, AZ 85721, USA. [2] Ronin Institute, Montclair, NJ 07043, USA. [3] Center for Genomics and Systems Biology, Department of Biology, New York University, New York, NY 10003, USA. [4] Department of Ecology and Evolutionary Biology, University of Arizona, Tucson, AZ 85721, USA. Correspondence and requests for materials should be addressed to J.M. (email: masel@email.arizona.edu)

Transcriptional regulatory networks (TRNs) are integral to development and physiology, and underlie all complex traits. An intriguing finding about TRNs is that certain topological "motifs" of interconnected transcription factors (TFs) are overrepresented relative to random re-wirings that preserve the frequency distribution of connections[1,2]. The significance of this finding remains open to debate.

The canonical example is the feed-forward loop (FFL), in which TF A regulates a target C both directly, and indirectly via TF B, and no regulatory connections exist in the opposite direction[1–3]. Each of the three regulatory interactions in an FFL can be either activating or repressing, so there are eight distinct kinds of FFLs (Supplementary Fig. 1)[4]. Given the eight frequencies expected from the ratio of activators to repressors, two of these kinds of FFLs are significantly overrepresented[4]. In this paper, we focus on one of these two overrepresented types, namely the type 1 coherent FFL (C1-FFL), in which all three links are activating rather than repressing (Supplementary Fig. 1, left). C1-FFL motifs are an active part of systems biology research today, e.g. they are used to infer the function of specific regulatory pathways[5,6].

The overrepresentation of FFLs in observed TRNs is normally explained in terms of selection favoring a function of FFLs. Specifically, the most common adaptive hypothesis is that cells often benefit from ignoring short-lived signals and responding only to durable signals[3,4,7]. Evidence that C1-FFLs can perform this function comes from the behavior both of theoretical models[4] and of in vivo gene circuits[7]. A C1-FFL can achieve this function when its regulatory logic is that of an AND gate, i.e. both the direct path from A to C and the indirect path from A to B to C must be activated before the response is triggered. In this case, the response will only be triggered if, by the time the signal trickles through the longer path, it is still active on the shorter path as well. This yields a response to long-lived signals but not short-lived signals.

However, just because a behavior is observed, we cannot conclude that the behavior is a historical consequence of past selection favoring that behavior[8,9]. The explanatory power of this adaptive hypothesis of filtering out short-lived and spurious signals needs to be compared to that of alternative, nonadaptive hypotheses[10]. The overrepresentation of C1-FFLs might be a byproduct of some other behavior that was the true target of selection[11]. Alternatively, it might be an intrinsic property of TRNs generated by mutational processes—gene duplication patterns have been found to enrich for FFLs in general[12], although not yet C1-FFLs in particular. Adaptationist claims about TRN organization have been accused of being just-so stories, with adaptive hypotheses still in need of testing against an appropriate null model of network evolution[13–23].

Here we develop such a computational null model of TRN evolution, and apply it to the case of C1-FFL overrepresentation. We include sufficient realism in our model of cis-regulatory evolution to capture the nonadaptive effects of mutation in shaping TRNs. In particular, we consider weak TF binding sites (TFBSs) that can easily appear de novo by chance alone, and from there be selected to bind a TF more strongly, as well as simulating mutations that duplicate and delete genes.

Our TRN model also captures the stochasticity of gene expression, which causes the number of mRNAs and hence proteins to fluctuate[24,25]. This is important, because demand for spurious signal filtering and hence C1-FFL function may arise not just from external signals, but also from internal fluctuations. Stochasticity in gene expression also shapes how external spurious signals are propagated. Stochasticity is a constraint on what TRNs can achieve, but it can also be adaptively co-opted in evolution[26]; either way, it might underlie the evolution of certain

motifs. Most other computational models of TRN evolution that consider gene expression as the major phenotype do not simulate stochasticity in gene expression (but see three notable exceptions[27–29]).

Given the potential importance of the details of stochastic and nonstochastic dynamics, we constrain the parameter ranges explored by mutation to values taken from data mostly on Saccharomyces cerevisiae. Different parameter values can cause the same network topology to display different dynamic behaviors; different topologies can also display similar dynamic behaviors[21,30–32].

Here we ask whether AND-gated C1-FFLs evolve as a response to selection for filtering out short and spurious external signals. Our new model allows us to compare the frequencies of network motifs arising in the presence of this hypothesized evolutionary cause to motif frequencies arising under nonadaptive control simulations, i.e. evolution under conditions that lack short spurious external signals while controlling both for mutational biases and for less specific forms of selection. We also ask whether other network motifs evolve to filter out short spurious signals, and if so, whether different conditions favor the appearance of different motifs during evolution.

## Results

**Model overview**. We simulate the dynamics of TRNs as the TFs activate and repress one another's transcription over a timescale we refer to as "gene expression time". This generates the gene expression phenotypes on which selection acts over longer evolutionary timescales. For each moment in gene expression time, we simulate the numbers of nuclear and cytoplasmic mRNAs in a cell, the protein concentrations, and the chromatin state of each gene in a haploid genome. Transitions between three possible chromatin states—Repressed, Intermediate, and Active—are a stochastic function of TF binding, and transcription initiation from the Active state is also stochastic. An overview of the model is shown in Fig. 1, with details given in the Methods. TF binding to the cis-regulatory sequence of a gene affects chromatin, which affects transcription rates, eventually feeding back to affect the concentration of TFs and hence their binding. Gene expression is further controlled by five gene-specific parameters: mean duration of transcriptional bursts, mRNA degradation rate, protein production rate, protein degradation rate, and gene length (which affects delays in transcription and translation).

We model five types of mutations: (1) to the five gene-specific parameters, (2) to the cis-regulatory sequences, (3) to the consensus binding sequences, (4) to the maximum binding affinity of TFs, and (5) duplication/deletion of genes. An external signal (Fig. 1a, red) is treated like another TF, and the concentration of an effector gene (Fig. 1a, blue) in response is a primary determinant of fitness, combined with a cost associated with gene expression (Fig. 1b). Mutants replace resident genotypes as a function of the difference in estimated fitness (Fig. 1c). Parameter values, taken as far as possible from S. cerevisiae, are summarized in Supplementary Table 1. Mutation rates are summarized in Supplementary Table 2.

C1-FFLs must be AND-gated to achieve their putative function. To allow the regulatory logic to evolve between AND-gated and other regulatory logic, we make effector gene expression require at least two TFBSs to be occupied by activators. An AND-gate is then present when the only way to have two TFs bound is for them to be different TFs (Fig. 2). All other genes are AND-gate-incapable, meaning that their activation requires only one TFBS to be occupied by an activator.

We select on the ability to recognize signals. In environment 1, expressing the effector is beneficial, and in environment 2 it is deleterious (see Methods for details). We select for TRNs that

**a**

Signal:

TF gene:

mRNA

Protein

Decay

Effector gene:

3-State transcription initiation

| Repressed (nucleosome bound) | Intermediate (nucleosome disassembled) | Active (transcriptional machinery bound) |

Transcriptional machinery

**b**

Concentration of effector protein — Gene expression time

Cost of gene expression — Gene expression time

Instantaneous fitness — Gene expression time

**c**

Discard mutant and restart with resident — No

Resident → Mutate in one of 5 ways → Mutant → Estimate mutant fitness $\hat{F}_{mutant}$ with 200 gene expression simulation replicates → $\dfrac{\hat{F}_{mutant} - \hat{F}_{resident}}{|\hat{F}_{resident}|} \geq 10^{-8}$?

Mutant replaces resident. Re-estimate new resident fitness $\hat{F}_{resident}$ with additional 800 gene expression simulation replicates — Yes

**Fig. 1** Overview of the model. **a** Simulation of gene expression phenotypes. We show a simple TRN with one TF (yellow) and one effector gene (blue), with arrows for major biological processes simulated in the model. **b** Phenotype–fitness relationship. Fitness is primarily determined by the concentration of an effector protein (here shown as beneficial as in Eq. 4, but potentially deleterious in a different environment as in Eq. 5), with a secondary component coming from the cost of gene expression (proportional to the rate of protein production), combined to give an instantaneous fitness at each moment in gene expression time. **c** Evolutionary simulation. A single resident genotype is replaced when a mutant's estimated fitness is high enough. Stochastic gene expression adds uncertainty to the estimated fitness, allowing less fit mutants to occasionally replace the resident, capturing the flavor of genetic drift

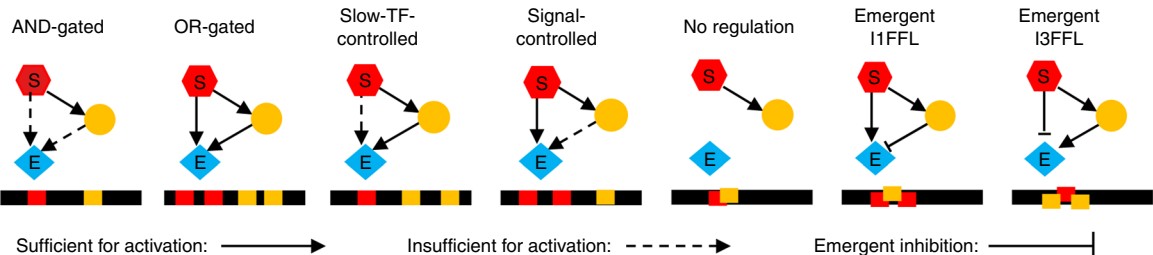

**Fig. 2** The distribution of TFBSs determines the regulatory logic of effector expression. We use the pattern of TFBSs (red and yellow bars along black *cis*-regulatory sequences) to classify the regulatory logic of the effector gene. C1-FFLs are classified first by whether or not they are capable of simultaneously binding the signal and the TF (left 4 vs. right 3; see Supplementary Fig. 2 and Supplementary Methods for details about overlapping TFBSs). Further classification is based on whether either the signal or the TF has multiple nonoverlapping TFBSs, allowing it to activate the effector without help from the other (solid arrow). The three subtypes to the right (where the signal and TF cannot bind simultaneously) are rarely seen; they are unless otherwise indicated included in "Any logic" and "non-AND-gated" tallies, but are not analyzed separately. Two of them involve emergent repression, creating incoherent feed-forward loops (see Supplementary Fig. 1 for full FFL naming scheme). Emergent repression occurs when the binding of one activator to its only TFBS prevents the other activator from binding to either of its two TFBSs, hence preventing simultaneous binding of two activators

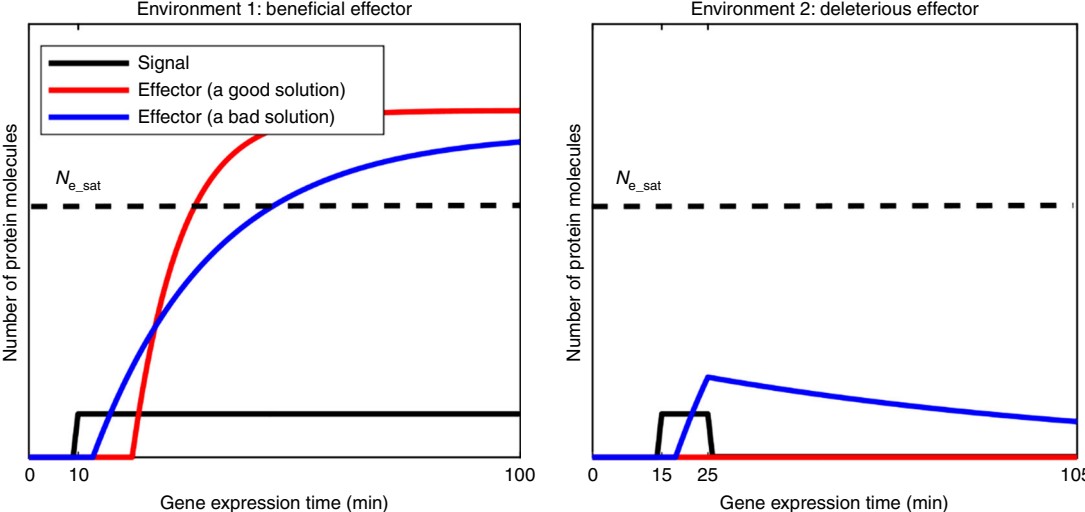

**Fig. 3** Selection for filtering out short spurious signals. Each selection condition averages fitness across simulations in two environments. The effectors have different fitness effects in the two environments, and the signal also behaves differently in the two environments. Simulations begin with zero mRNA and protein, and all genes at the Repressed state (see Methods). Each simulation is burned in for a randomly sampled length of time in the absence of signal (shown here as 10 min in environment 1, and 15 min in environment 2), and continues for another 90 min after the burn-in. The signal is shown in black. Red illustrates a good solution in which the effector responds appropriately in each of the environments, while blue shows an inferior solution. $N_{e\_sat}$ marks the amount of effector protein at which the benefit from expressing the effector in environment 1 becomes saturated, as does the damage in environment 2 (see Methods). See Supplementary Fig. 3 for examples of high-fitness and low-fitness evolved phenotypes, where, as shown in this schematic, high-fitness solutions have longer delays followed by more rapid responses thereafter

take information from the signal and correctly decide whether to express the effector. Fitness is a weighted average across separate gene expression simulations in the two environments and their corresponding presence or absence of signal. In both cases, we begin each gene expression simulation with no signal. In both environments, the signal is turned on after a burn-in period (see Methods), but in environment 2, the signal lasts for only 10 min, with selection to ignore it (Fig. 3).

**AND-gated C1-FFLs readily evolve as spurious signal filters**. We begin by simulating the easiest case we can devise to allow the evolution of C1-FFLs for their purported function of filtering out short spurious signals. The signal is allowed to act directly on the effector, after which all that needs to evolve is a single activating TF between the two, as well as AND-logic for the effector (Fig. 2, leftmost). We score network motifs at the end of a set period of evolution (see Supplementary Methods for details), further classifying evolved C1-FFLs into subtypes based on the presence of nonoverlapping TFBSs (Fig. 2). The adaptive hypothesis predicts the evolution of the C1-FFL subtype with AND-regulatory logic, which requires the effector to be stimulated both by the signal and by the slow TF. While all evolutionary replicates show large increases in fitness, the extent of improvement varies dramatically, indicating whether or not the replicate was successful at evolving the phenotype of interest rather than becoming stuck at an alternative locally optimal phenotype (Fig. 4a). AND-gated C1-FFLs frequently evolve in replicates that reach high fitness, but not in replicates that reach lower fitness (Fig. 4b).

We also see C1-FFLs that, contrary to expectations, are not AND-gated. Non-AND-gated motifs are found more often in low-fitness than high-fitness replicates (Fig. 4b), indicating that the preference for AND-gates is associated with adaptation rather than mutation bias. However, some non-AND-gated motifs are still found even in the high-fitness replicates. This is because motifs and their logic gates are scored on the basis of all TFBSs, even those with two mismatches and hence low binding affinity. Unless these weak TFBSs are deleterious, they will appear quite

often by chance alone. A random 8-bp sequence has probability $\binom{8}{2} \times 0.25^6 \times 0.75^2 = 0.0038$ of being a two-mismatch binding site for a given TF. In our model, a TF has the potential to recognize 137 different sites in a 150-bp *cis*-regulatory sequence (taking into account steric hindrance at the edges), each with two orientations. Thus, by chance alone a given TF will have $0.0038 \times 137 \times 2 \approx 1$ two-mismatch binding sites in a given *cis*-regulatory sequence (ignoring palindromes for simplicity), compared to only ~0.1 one-mismatch TFBSs. Non-AND-gated C1-FFLs mostly disappear when two-mismatch TFBSs are excluded, but the AND-gated C1-FFLs found in high-fitness replicates do not (Fig. 4c).

To confirm the functionality of these AND-gated C1-FFLs, we mutated the evolved genotype in two different ways (Fig. 5a) to remove the AND regulatory logic. As expected, this lowers fitness in the presence of the short spurious signal but increases fitness in the presence of constant signal, with a net reduction in fitness (Fig. 5b). This is consistent with AND-gated C1-FFLs representing a tradeoff, by which a more rapid response to a true signal is sacrificed in favor of the greater reliability of filtering out short spurious signals.

Adaptive motifs are constrained not only in their topology and regulatory logic, but also in the parameter space of their component genes. In particular, there is selection for rapid synthesis of both effector and TF proteins, as well as rapid degradation of effector mRNA and protein (Supplementary Table 3). Fast effector degradation reduces the transient expression induced by the short spurious signal (Supplementary Fig. 3). Note that we evolved solutions only at the level of transcriptional regulation—even more rapid switching could be achieved by posttranslational modifications, if we had allowed them in our model.

To test the extent to which AND-gated C1-FFLs are a specific response to selection to filter out short spurious signals, we simulated evolution under three negative control conditions: (1) no selection, i.e. all mutations are accepted to become the new

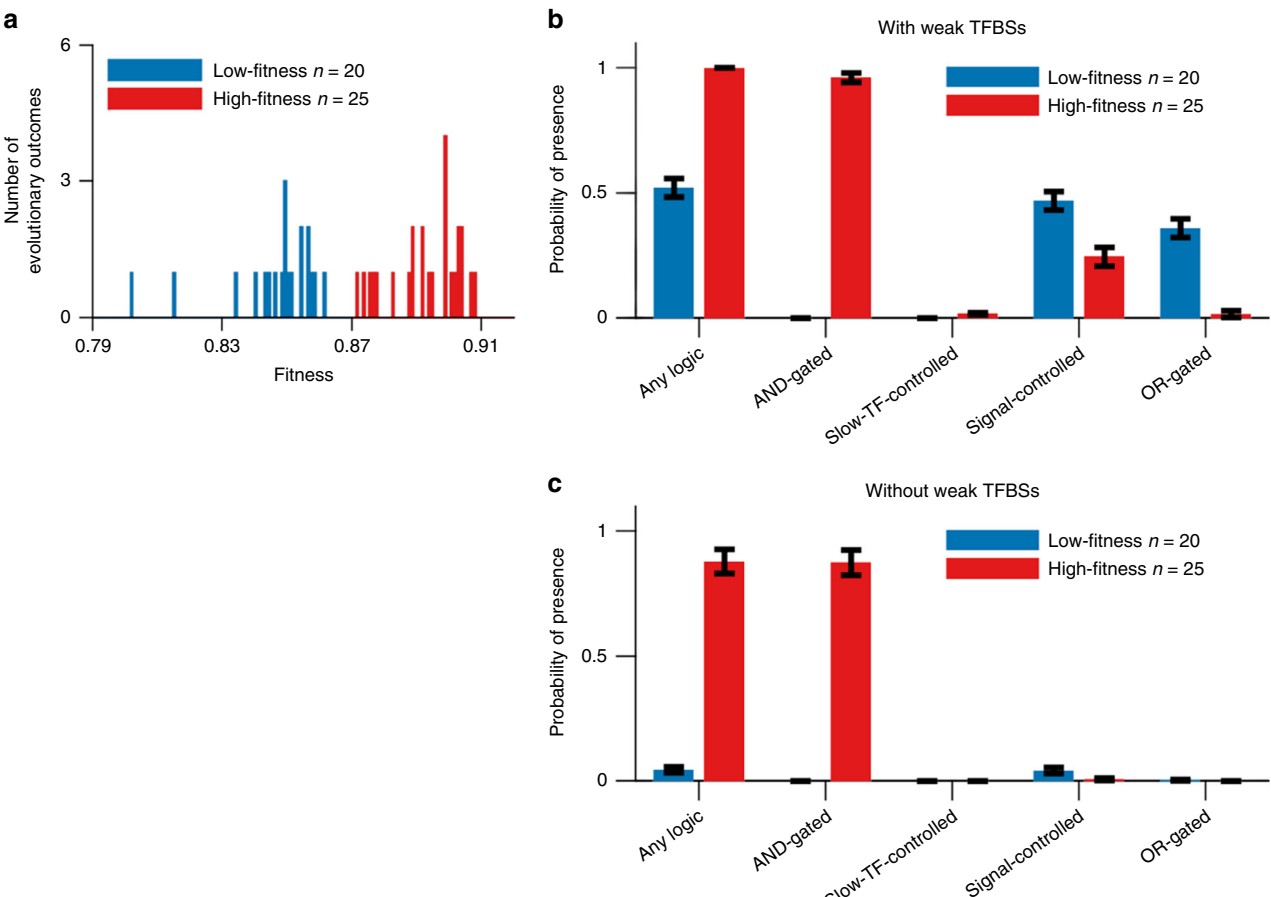

**Fig. 4** AND-gated C1-FFLs are associated with a successful response to selection. **a** Distribution of fitness outcomes across replicate simulations, calculated as the average fitness over the last 10,000 steps of the evolutionary simulation. We divide genotypes into a low-fitness group (blue) and a high-fitness group (red) using as a threshold an observed gap in the distribution. **b** High-fitness replicates are characterized by the presence of an AND-gated C1-FFL. "Any logic" counts the presence of any of the seven subtypes shown in Fig. 2b. Because one TRN can contain multiple C1-FFLs of different subtypes, each of which are scored, the sum of the occurrences of all seven subtypes will generally be more than "Any logic". See Supplementary Methods for details on the calculation of the y axis. **c** The overrepresentation of AND-gated C1-FFLs becomes even more pronounced relative to alternative logic-gating when weak (two-mismatch) TFBSs are excluded while scoring motifs. Data are shown as mean ± s.e.m. of the occurrence over replicate evolution simulations

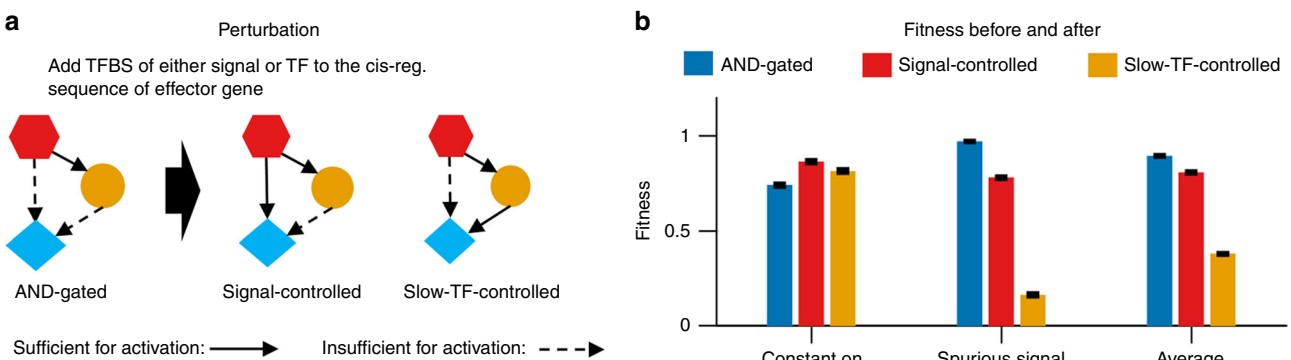

**Fig. 5** Destroying the AND-logic of a C1-FFL removes its ability to filter out short spurious signals. **a** For each of the $n = 25$ replicates in the high-fitness group in Fig. 4, we perturbed the AND-logic in two ways, by adding one binding site of either the signal or the slow TF to the *cis*-regulatory sequence of the effector gene. **b** For each replicate, the fitness of the original motif (blue) or of the perturbed motif (red or orange) was averaged across the subset of evolutionary steps with an AND-gated C1-FFL and lacking other potentially confounding motifs (see Supplementary Fig. 4 and Supplementary Methods for details). Destroying the AND-logic slightly increases the ability to respond to the signal, but leads to a larger loss of fitness when short spurious signals are responded to. Fitness is shown as mean ± s.e.m. over replicate evolutionary simulations

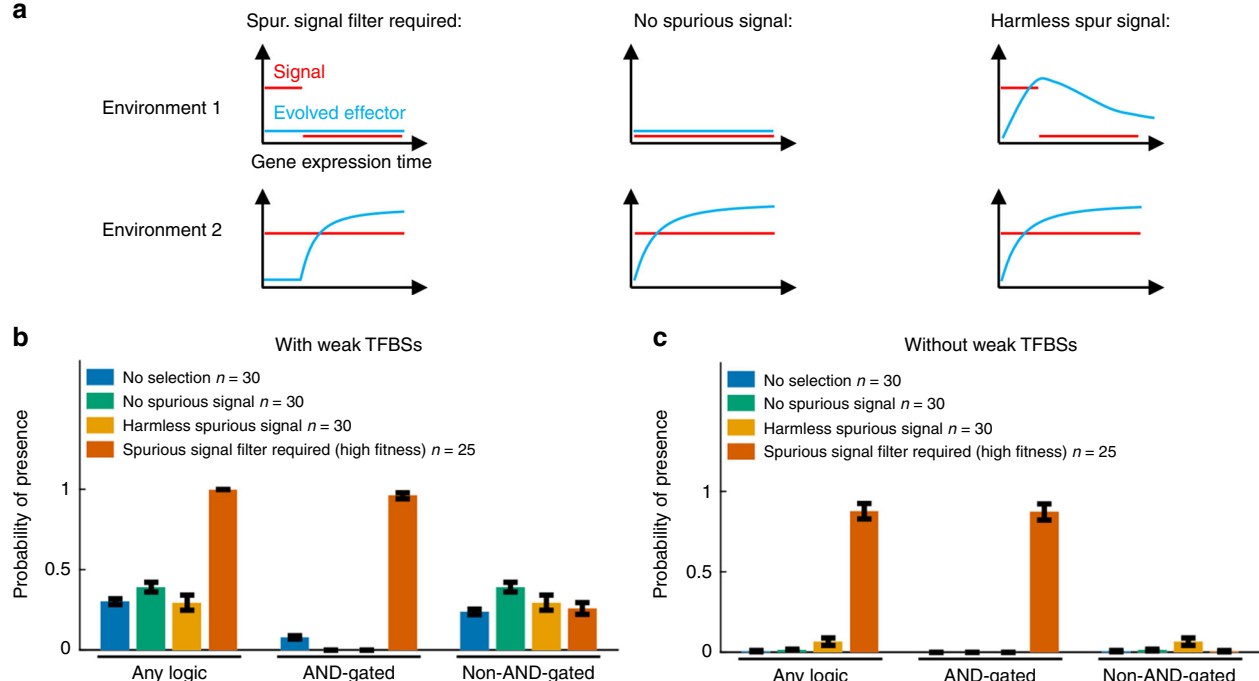

**Fig. 6** Selection for filtering out short spurious signals is the primary cause of C1-FFLs. TRNs are evolved under different selection conditions, and we score the probability that at least one C1-FFL is present (see Supplementary Methods). Schematics of selection, in which fitness is averaged with weights 2:1 over environment 1:2, are shown in (**a**). The effector is deleterious in environment 1 except in the "harmless" and "no selection" conditions. Weak (two-mismatch) TFBSs are included in (**b**) and are excluded in (**c**) during motif scoring. Data are shown as mean ± s.e.m. over evolutionary replicates. C1-FFL occurrence is similar for high-fitness and low-fitness outcomes in control selective conditions (Supplementary Fig. 5), and so all evolutionary outcomes were combined. "Spurious signal filter required (high fitness)" uses the same data as in Fig. 4

resident genotype; (2) no spurious signal, i.e. selection to express the effector under a constant ON signal and not under a constant OFF signal (Fig. 6a); (3) harmless spurious signal, i.e. selection to express the effector under a constant ON signal whereas effector expression in the OFF environment with short spurious signals is neither punished nor rewarded beyond the cost of unnecessary gene expression (Fig. 6a). AND-gated C1-FFLs evolve much less often under all three negative control conditions (Fig. 6b, c), showing that their prevalence is a consequence of selection for filtering out short spurious signals, rather than a consequence of mutational bias and/or simpler forms of selection. C1-FFLs that do evolve under control conditions tend not to be AND-gated (Fig. 6b), and mostly disappear when weak TFBSs are excluded during motif scoring (Fig. 6c).

**More complex networks also evolve diamond motifs**. In real biological situations, sometimes the source signal will not be able to directly regulate an effector, and must instead operate via a longer regulatory pathway involving intermediate TFs[33]. In this case, even if the signal itself takes the idealized form shown in Fig. 3, its shape after propagation may become distorted by the intrinsic processes of transcription. Motifs are under selection to handle this distortion.

To enforce indirect regulation, we ran simulations in which the signal was only allowed to bind to the *cis*-regulatory sequences of TFs and not of effector genes. The fitness distribution of the evolutionary replicates has no obvious gaps (Supplementary Fig. 6), so we compared the highest fitness, lowest fitness, and median fitness replicates. In agreement with results when direct regulation is allowed, genotypes of low and medium fitness contain few AND-gated C1-FFLs, while high-fitness genotypes contain many more (Fig. 7b, left and right).

While visually examining the network context of these C1-FFLs, we discovered that many were embedded within AND-gated "diamonds". In a diamond, the signal activates the expression of two genes that encode different TFs, and the two TFs activate the expression of an effector gene (Fig. 7a middle). When one of the two TF genes activates the other, then a C1-FFL is also present among the same set of genes; we call this topology a "FFL-in-diamond" (Fig. 7a, right), and the prevalence of this configuration drew our attention toward diamonds. This led us to discover that AND-gated diamonds also occurred frequently without AND-gated C1-FFLs, in the configuration we call "isolated diamonds" (Fig. 7a, middle). Note that it is in theory possible, but in practice uncommon, for diamonds to be part of more complex conjugates. Systematically scoring the AND-gated isolated diamond motif confirmed its high occurrence (Fig. 7b, c, middle). AND-gated isolated C1-FFLs appear mainly in the highest fitness outcomes, while AND-gated isolated diamonds appear in all fitness groups (Fig. 7c), suggesting that diamonds are easier to evolve.

An AND-gated C1-FFL integrates information from a short/fast regulatory pathway with information from a long/slow pathway, in order to filter out short spurious signals. A diamond achieves the same end of integrating fast and slowly transmitted information via differences in the gene expression dynamics of the two regulatory pathways, rather than via topological length (Fig. 8). The fast and slow pathways could be distinguished in a number of ways, e.g. by the slope at which the transcription factor concentration increases or the time at which it exceeds a threshold or plateaus. We found it convenient to identify the "fast TF" as the one with the higher protein degradation rate. Specifically, we use the geometric mean of the protein degradation rate over gene copies of a TF in order to differentiate the two TFs.

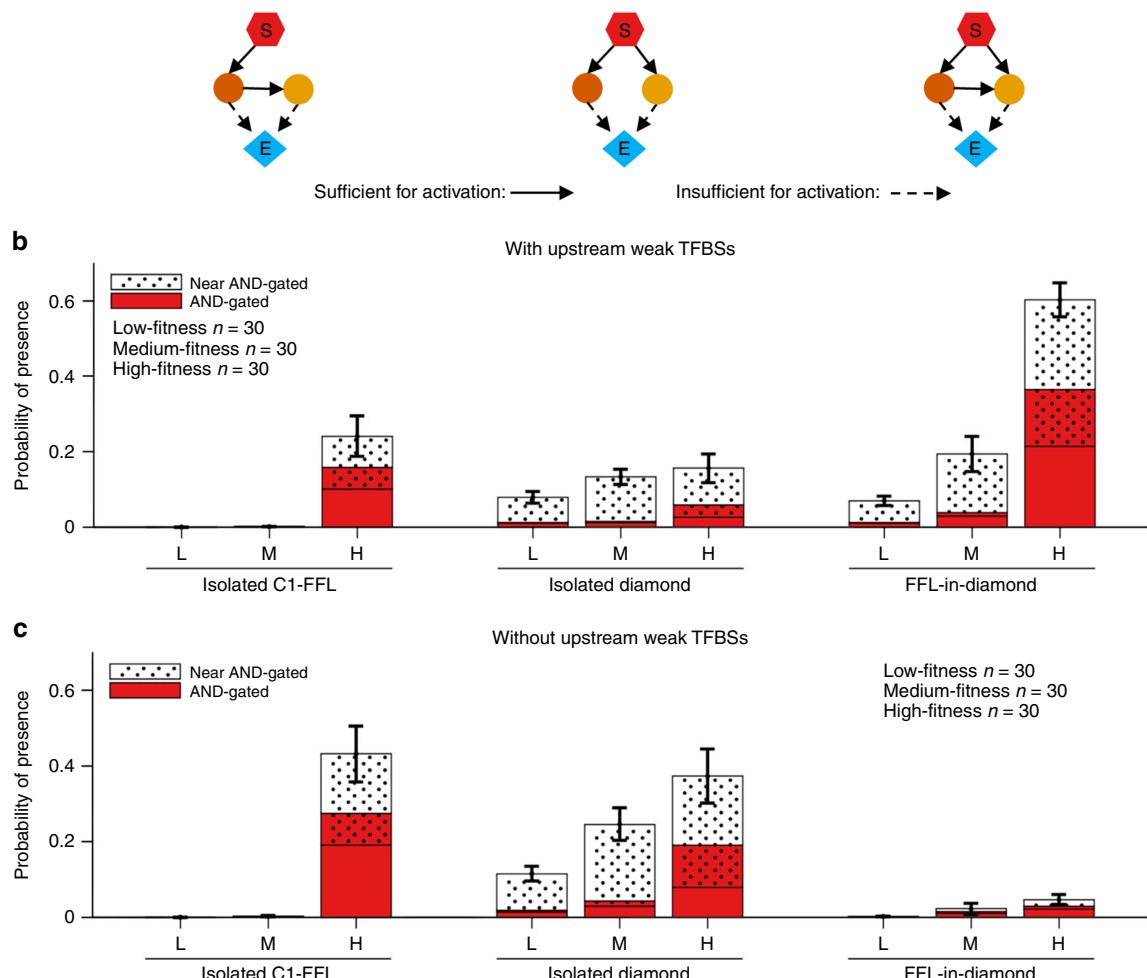

**Fig. 7** AND-gated C1-FFLs and diamonds are associated with high fitness in complex networks. Out of 238 simulations (Supplementary Fig. 6), we took the 30 with the highest fitness (H), the 30 with the lowest fitness (L), and 30 of around median fitness (M). AND-gated motifs are scored while including weak TFBSs in the effectors' *cis*-regulatory regions, near-AND-gated motifs are those scored only when these weak TFBSs are excluded. **a** Diagrams of enriched motifs when weak TFBSs are included. It is possible for the same genotype to contain one of each, resulting in overlap between the red AND-gated columns and the dotted near-AND-gated columns. Weak TFBSs upstream in the TRN, i.e. not in the effector, are shown both included (**b**) and excluded (**c**). See Supplementary Methods for *y*-axis calculation details. Error bars show mean ± s.e.m. of the proportion of evolutionary steps containing the motif in question, across replicate evolutionary simulations

The parameter values of the fast TF are more evolutionarily constrained than those of the slow TF (Supplementary Table 4). In particular, there is selection for rapid degradation of the fast TF protein and mRNA (Supplementary Table 4). Isolated AND-gated C1-FFLs also show pronounced selection for the TF in the fast pathway to have rapid protein degradation (Supplementary Table 5). Fast-degrading mRNA and proteins are rare (Supplementary Table 1), suggesting that mutational biases might make them difficult to evolve. But even when they do evolve, fast degradation keeps the fast TF at low concentrations. To compensate, the fast TF must overcome mutational bias to also evolve high binding affinity and rapid protein synthesis (Supplementary Tables 4 and 5).

Perturbation analysis supports an adaptive function for AND-gated C1-FFLs and diamonds evolved under indirect regulation (Fig. 9a, b). Breaking the AND-gate logic of these motifs by adding a TFBS to the effector *cis*-regulatory region reduces the fitness under the spurious signal but increases it under the constant ON signal, resulting in a net decrease in the overall fitness. Note that a simple transcriptional cascade, signal → TF → effector, has also

been found experimentally to filter out short spurious signals[34]; in Supplementary Note 1, we argue that diamonds are not by-products of selection for cascades, but are the direct target of selection.

**Weak TFBSs change how motifs are scored**. Results depend on whether we include weak TFBSs when scoring motifs. Weak TFBSs can either be in the effector's *cis*-regulatory region, affecting how the regulatory logic is scored, or in TFs upstream in the TRN, affecting only the presence or absence of motifs.

When we exclude upstream weak TFBSs while scoring motifs, FFL-in-diamonds are no longer found, while the occurrence of isolated C1-FFLs and diamonds increases (Fig. 7c). This makes sense, because adding one weak TFBS, which can easily happen by chance alone, can convert an isolated diamond or C1-FFL into a FFL-in-diamond (added between intermediate TFs, or from signal to slow TF, respectively).

When a motif is scored as AND-gated only when two-mismatch TFBSs in the effector are excluded, we call it a "near-AND-gated" motif. TFs may bind so rarely to a weak affinity

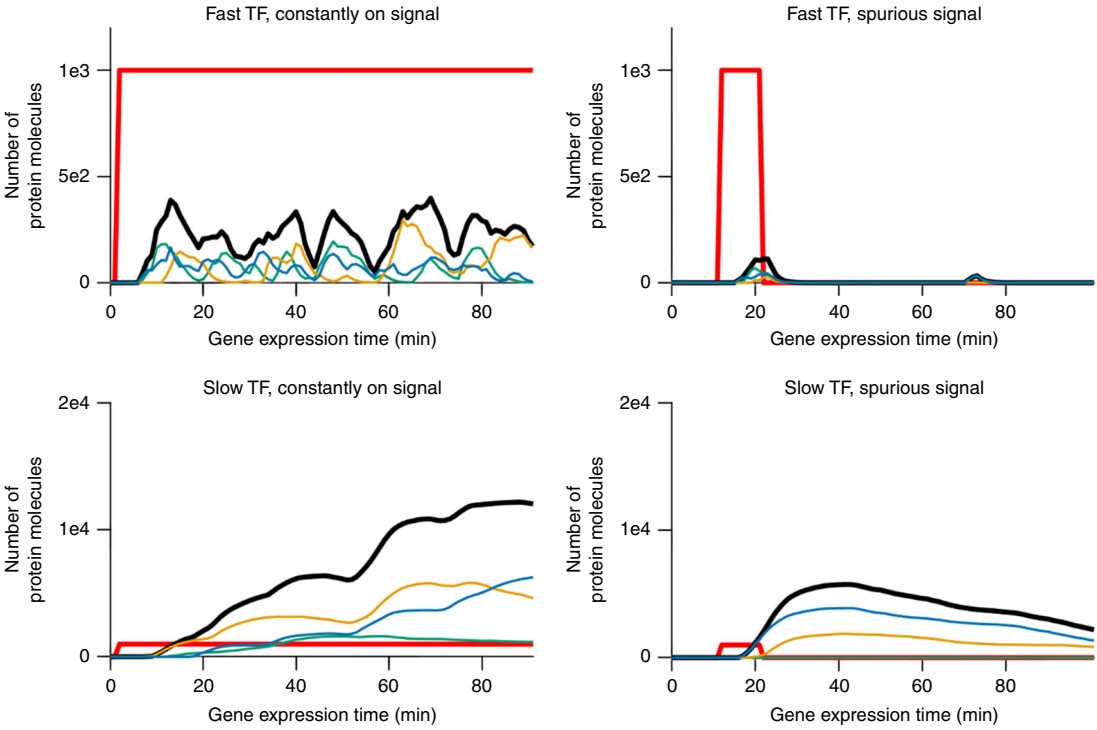

**Fig. 8** The two TFs in an AND-gated diamond propagate the signal at different speeds. Expression of the two TFs in one representative genotype from the one high-fitness evolutionary replicate in Fig. 7b that evolved an AND-gated isolated diamond is shown. Both the slow TF and the fast TF are encoded by three gene copies, shown separately in color, with the total for each TF in thick black. The expression of one TF plateaus faster than that of the other; this is characteristic of the AND-gated diamond motif, and leads to the same functionality as the AND-gated C1-FFL

TFBS that the presence of the weak TFBS changes little, making the regulatory logic still effectively AND-gated. A near-AND-gated motif may therefore evolve for the same adaptive reasons as an AND-gated one. Figure 7b, c shows that both AND-gated and near-AND-gated motifs are enriched in the higher fitness genotypes. There is more likely to be a weak affinity TFBS for the fast TF than the slow TF, and adding one does less harm (see Supplementary Note 2 and Supplementary Fig. 7).

**Diamonds also evolve without external spurious signals.** We simulated evolution under the same three control conditions as before, this time without allowing the signal to directly regulate the effector. When weak (two-mismatch) TFBSs are excluded, AND-gated isolated C1-FFLs are seen only after selection for filtering out a spurious signal, and not under other selection conditions (Fig. 10a). However, AND-gated isolated diamonds also evolve in the absence of spurious signals, indeed at even higher frequency (Fig. 10b). Results including weak TFBSs are similar (Supplementary Fig. 10).

Perturbing the AND-gate logic in isolated diamonds evolved in the absence of spurious external signals reduces fitness via effects in the environment where expressing the effector is deleterious (Fig. 9c). The stochastic expression of intermediate TFs might effectively create short intrinsic spurious signals when the external signal is set to OFF. It seems that AND-gated diamonds evolve to mitigate this risk, but that AND-gated C1-FFLs do not. This may be because C1-FFLs delay the expression of effectors more than diamonds do, because the fast TF must first be translated in order to turn on the slow TF. Delays are costly when expression is beneficial, and unnecessary to filter out a very short signal; because internally generated spurious signals have an exponential distribution, most are short[35]. Alternatively, the advantage of diamonds might be that spurious effector expression requires both TFs to be accidentally and independently expressed,

whereas spurious TF expression in AND-gated C1-FFLs is not independent because the fast TF can induce the slow TF[36].

## Discussion

There has never been sufficient evidence to satisfy evolutionary biologists that motifs in TRNs represent adaptations for particular functions. Critiques by evolutionary biologists to this effect[13–23] have been neglected, rather than answered, until now. While C1-FFLs can be conserved across different species[37–40], this does not imply that specific "just-so" stories about their function are correct. In this work, we study the evolution of AND-gated C1-FFLs, which are hypothesized to be adaptations for filtering out short spurious signals[3]. Using a novel and more mechanistic computational model to simulate TRN evolution, we found that AND-gated C1-FFLs evolve readily under selection for filtering out a short spurious signal, and not under control conditions. Our results support the adaptive hypothesis about C1-FFLs.

It is difficult to distinguish adaptations from "spandrels"[8]. Standard procedure is to look for motifs that are more frequent than expected from some randomized version of a TRN[2,41]. For this method to work, this randomization must control for all confounding factors that are nonadaptive with respect to the function in question, from patterns of mutation to a general tendency to hierarchy—a near-impossible task. Our approach to a null model is not to randomize, but to evolve with and without selection for the specific function of interest. This meets the standards of evolutionary biology for inferring the adaptive nature of a motif[13–23].

AND-gated C1-FFLs express an effector after a noise-filtering delay when the signal is turned on, but shut down expression immediately when the signal is turned off, giving rise to a "sign-sensitive delay"[3,7]. Rapidly switching off has been hypothesized to be part of their selective advantage, above and beyond the

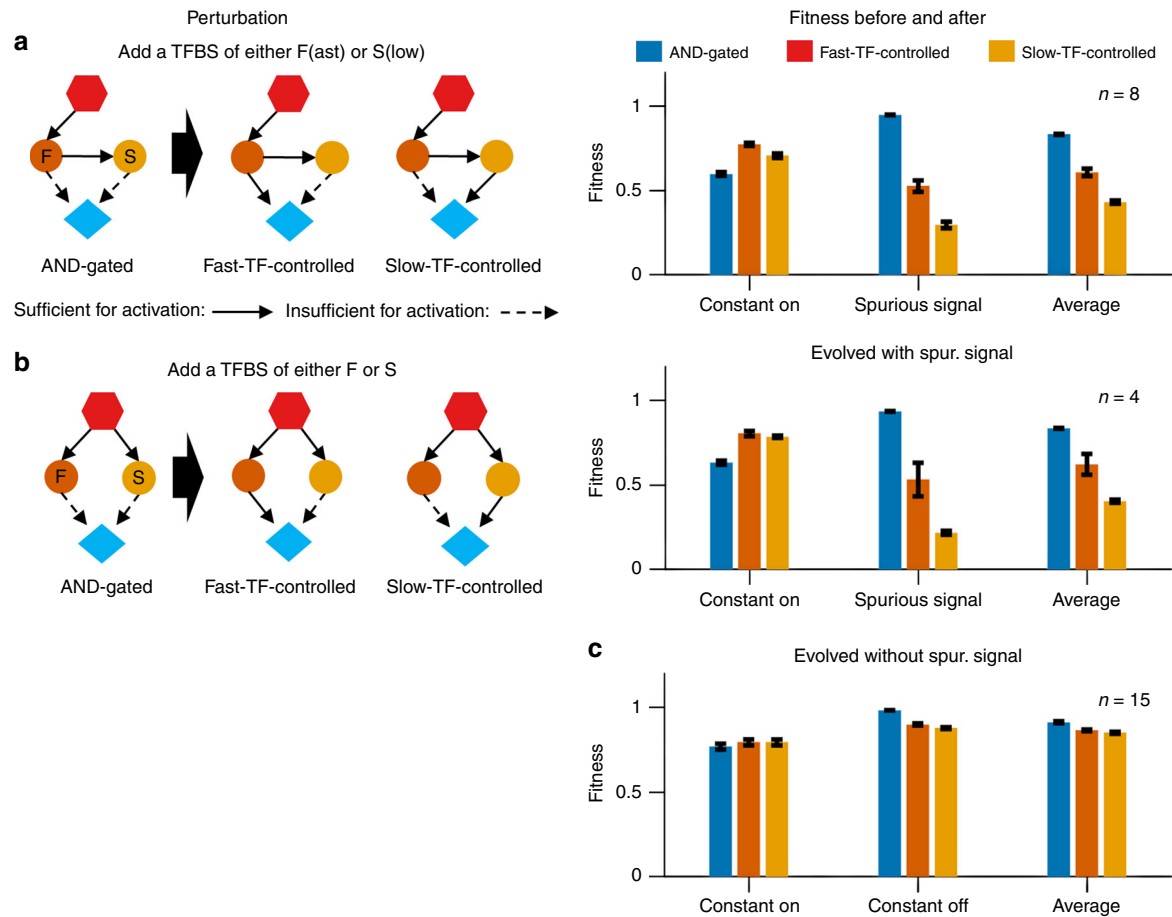

**Fig. 9** Isolated C1-FFLs and diamonds rely on AND gates to filter out short spurious signals. We add a TFBS of either the fast TF or the slow TF to break the AND gate. This slightly increases the ability to respond to the signal, but leads to a larger loss of fitness when effector expression is undesirable. We perform the perturbation on **a** 8 of the 18 high-fitness replicates from Fig. 7b that evolved an AND-gated C1-FFL, **b** 4 of the 26 high-fitness replicates that evolved an AND-gated diamond in Fig. 7b, and **c** 15 of the 37 replicates that evolved an AND-gated diamond in response to selection for signal recognition in the absence of an external spurious signal (Fig. 10b). Replicate exclusion was based on the co-occurrence of other motifs with the potential to confound results (see Supplementary Methods for details). Fitness is shown as mean ± s.e.m. of over replicate evolutionary simulations, calculated as described for Fig. 5

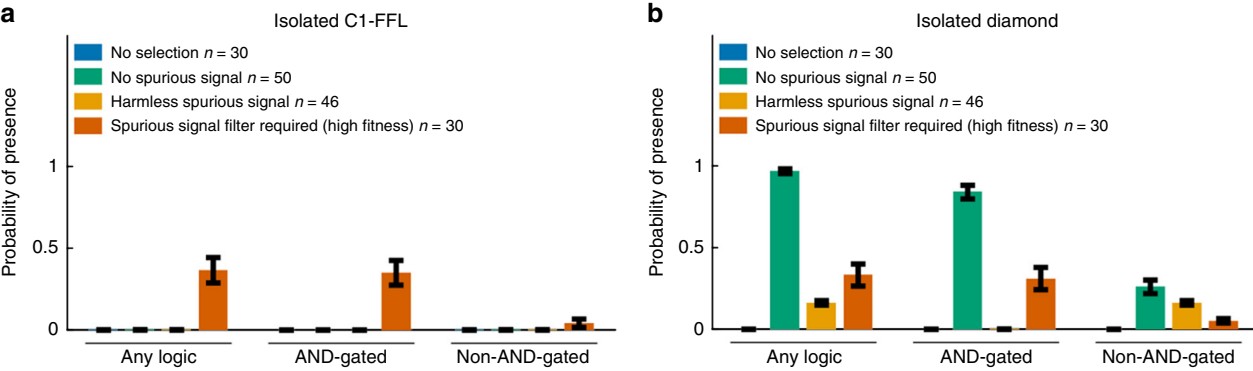

**Fig. 10** AND-gated diamonds, but not AND-gated C1-FFLs, also evolve in negative controls. **a** Selection for filtering out a short spurious signal is the primary way to evolve AND-gated isolated C1-FFLs, but **b** AND-gated isolated diamonds also evolve in the absence of spurious signals. The selection conditions are the same as in Fig. 6, but we do not allow the signal to directly regulate the effector. In the "no spurious signal" and "harmless spurious signal" control conditions, motif frequencies are similar between low- and high-fitness genotypes (Supplementary Figs. 8 and 9), and so our analysis includes all evolutionary replicates. When scoring motifs, we exclude all two-mismatch TFBSs; results with inclusions, and for FFL-in-diamonds, are shown in Supplementary Fig. 10. Many non-AND-gated diamonds have the "no regulation" logic in Fig. 2, perhaps as an artifact created by the duplication and divergence of intermediate TFs; we excluded them from the "Any logic" and "Non-AND-gated" tallies in (**b**). See Supplementary Methods for the calculation of the y-axis. Data are shown as mean ± s.e.m. over evolutionary replicates. We reused data from Fig. 7 for "Spurious signal filter required (high fitness)"

function of filtering out short spurious signals[35]. We intended to select only for filtering out a short spurious signal, and not for fast turn-off; specifically, we expected effector expression to evolve a delay equal to the duration of the spurious signal. However, evolved solutions still expressed the effector in the presence of short spurious signals (Supplementary Fig. 3), and thus benefitted from rapidly turning off this spurious expression. In other words, we effectively selected for both delayed turn-on and rapid turn-off, despite our intent to only select for the former.

Previous studies have also attempted to evolve adaptive motifs in a computational TRN, successfully under selection for circadian rhythm and for multiple steady states[42], and unsuccessfully under selection to produce a sine wave in response to a periodic pulse[23]. Other studies have evolved adaptive motifs in a mixed network of transcriptional regulation and protein–protein interaction[43–45]. Our successful simulation might offer some methodological lessons, especially a focus on high-fitness evolutionary replicates, which was done by us and by Burda et al.[42] but not by Knabe et al.[23].

Knabe et al.[23] suggested that including a cost for gene expression may suppress unnecessary links and thus make it easier to score motifs. However, when we removed the cost of gene expression term ($C(t) = 0$, see Methods), AND-gated C1-FFLs still evolved in the high-fitness genotypes under selection for filtering out a spurious signal (Supplementary Fig. 11). In our model, removing the cost of gene expression did not, via permitting unnecessary links, conceal motifs.

While simplified relative to reality, our model is undeniably complicated. An important question is which complications are important for what. One complication is our nucleotide-sequence-level model of *cis*-regulatory sequences. This has the advantage of capturing weak TFBSs, realistic turnover, and other mutational biases. The disadvantage is that calculating the probabilities of TF binding is computationally expensive and scales badly with network size. Future work might design a more schematic model of *cis*-regulatory sequences to improve computation while still capturing realistic mutation biases. A second complication of our approach is the stochastic simulation of gene expression. This is essential for our question, because intrinsic noise in gene expression can mimic the effects of a spurious signal, but may be less important in other scenarios in which the focus is on steady state behavior.

Our model, while complex for a model and hence capable of capturing intrinsic noise, is inevitably less complex than the biological reality. However, we hope to have captured key phenomena, albeit in simplified form. One key phenomenon is that TFBSs are not simply present vs. absent but can be strong or weak, i.e. the TRN is not just a directed graph, but its connections vary in strength. Our model, like that of Burda et al.[42] in the context of circadian rhythms, captures this fact by basing TF binding affinity on the number of mismatch deviations from a consensus TFBS sequence. While in reality, the strength of TF binding is determined by additional factors, such as broader nucleic context and cooperative behavior between TFs (reviewed in Inukai et al.[46]), these complications are unlikely to change the basic dynamics of frequent appearance of weak TFBSs and greater mutational accessibility of strong TFBSs from weak TFBSs than de novo. Similarly, AND-gating can be quantitative rather than qualitative[47], a phenomenon that weak TFBSs in our model provide a simplified version of.

Core links in adaptive motifs almost always involve strong not weak TFBSs. However, weak (two-mismatch) TFBSs can create additional links that prevent an adaptive motif from being scored as such. Some potential additional links are neutral while others are deleterious; the observed links are thus shaped by this selective filter, without being adaptive. Note that there have been experimental reports that even weak TFBSs can be functionally important[48,49]; these might, however, better correspond to 1-mismatch TFBSs in our model than two-mismatch TFBSs. Ramos and Barolo[49] and Crocker et al.[48] identified their "weak" TFBSs in comparison to the strongest possible TFBS, not in comparison to the weakest still showing affinity above baseline.

A striking and unexpected finding of our study was that AND-gated diamonds evolved as an alternative motif for filtering out short spurious external signals, and that these, unlike FFLs, were also effective at filtering out intrinsic noise. Multiple motifs have previously been found capable of generating the same steady state expression pattern[21]; here we find multiple motifs for a much more complex function.

Diamonds are not overrepresented in the TRNs of bacteria[2] or yeast[50], but are overrepresented in signaling networks (in which posttranslational modification plays a larger role)[51], and in neuronal networks[1]. In our model, we treated the external signal as though it were a transcription factor, simply as a matter of modeling convenience. In reality, signals external to a TRN are by definition not TFs (although they might be modifiers of TFs). This means that our indirect regulation case, in which the signal is not allowed to directly turn on the effector, is the most appropriate one to analyze if our interest is in TRN motifs that mediate contact between the two. Note that if under indirect regulation we were to score the signal as not itself a TF, we would observe adaptive C1-FFLs but not diamonds, in agreement with the TRN data. However, these TRN data might miss functional diamond motifs that spanned levels of regulatory organization, i.e. that included both transcriptional and other forms of regulation. The greatest chance of finding diamonds within TRNs alone come from complex and multilayered developmental cascades, rather than bacterial or yeast[52]. Multiple interwoven diamonds are hypothesized to be embedded with multilayer perceptrons that are adaptations for complex computation in signaling networks[53].

Previous work has also identified alternatives to AND-gated C1-FFLs. Specifically, in mixed networks of transcriptional regulation and protein−protein interactions, FFLs did not evolve under selection for delayed turn-on (as well as rapid turn-off)[45]. Indeed, even when an FFL topology was enforced, with only the parameters allowed to evolve, two alternative motifs remained superior[45]. However, one alternative motif, which the authors called "positive feedback" is essentially still an AND-gated C1-FFL, specifically one in which the intermediate TF expression is also AND-gated, requiring both itself and the signal for upregulation. The other is a cascade in which the signal inhibits the expression of an intermediate TF protein that represses the expression of the effector. The cost of constitutive expression of the intermediate TF in the absence of the signal was not modeled[45], giving this cascade an unrealistic advantage.

Most previous research on C1-FFLs has used an idealized implementation (e.g. a square wave) of what a short spurious signal entails[4,35,54]. In real networks, noise arises intrinsically in a greater diversity of forms, which our model does more to capture. Even when a "clean" form of noise enters a TRN, it subsequently gets distorted with the addition of intrinsic noise[55]. Intrinsic noise is ubiquitous and dealing with it is an omnipresent challenge for selection. Indeed, we see adaptive diamonds evolve to suppress intrinsic noise, even when we select in the absence of extrinsic spurious signals.

The function of a motif relies ultimately on its dynamic behavior, with topology merely a means to that end. To create two pathways that regulate the effector at different speeds, the C1-FFL motif uses a pair of short and long pathways, but these also correspond to fast-degrading and slow-degrading TFs. This same function was achieved entirely nontopologically in our

adaptively evolved diamond motifs. This agrees with other studies showing that topology alone is not enough to infer activities such as spurious signal filtering from network motifs[30–32].

## Methods

**Transcription factor binding.** Transcription of each gene is controlled by TFBSs present within a 150-bp *cis*-regulatory region. When bound, a TF occupies a stretch of DNA 14 bp long (Supplementary Fig. 2). In the center of this stretch, each TF recognizes an 8-bp consensus sequence (Supplementary Fig. 2), and binds to it with a TF-specific (and mutable) dissociation constant $K_d(0)$. TFs also bind somewhat specifically when there are one or two mismatches, with $K_d(1)$ and $K_d(2)$ values calculated from $K_d(0)$ according to a model of approximately additive binding energy per base pair. With three mismatches, binding occurs at the same background affinity as to any 14 bp stretch of DNA. We model competition between a smaller number of specific higher-affinity binding sites and the much larger number of nonspecific binding sites, the latter corresponding to the total amount of nucleosome-free sequence in *S. cerevisiae*. Competition with nonspecific binding can be approximated by using an effective dissociation constant $\hat{K}_d = 10K_d$. See Supplementary Methods for justification and details of these model choices.

Each TF is either an activator or a repressor. The algorithm for obtaining the probability distribution for $A$ activators and $R$ repressors being bound to a given *cis*-regulatory region at a given moment in gene expression time is described in the Supplementary Methods.

The signal is treated as though it were an activating TF whose concentration is controlled externally, with an OFF concentration of zero and an ON concentration of 1000 molecules per cell, which is the typical per-cell number of a yeast TF[56].

**Transcriptional regulation.** $P_A$ denotes the probability of having at least one activator bound for an AND-gate-incapable gene, or two for an AND-gate-capable gene. $P_R$ denotes the probability of having at least one repressor bound.

Noise in yeast gene expression is well described by a two-step process of transcriptional activation[57,58], e.g. nucleosome disassembly followed by transcription machinery assembly. We denote the three corresponding possible states of the transcription start site as Repressed, Intermediate, and Active (Fig. 1a). Transitions between the states depend on the numbers of activator and repressor TFs bound (e.g. via recruitment of histone-modifying enzymes[59,60]). We make conversion from Repressed to Intermediate a linear function of $P_A$, ranging from the background rate 0.15 min$^{-1}$ of histone acetylation[61] (presumed to be followed by nucleosome disassembly), to the rate of nucleosome disassembly 0.92 min$^{-1}$ for the constitutively active *PHO5* promoter[57]:

$$r_{\text{Rep\_to\_Int}} = 0.92P_A + 0.15(1 - P_A). \tag{1}$$

We make conversion from Intermediate to Repressed a linear function of $P_R$, ranging from a background histone de-acetylation rate of 0.67 per min[61], up to a maximum of 4.11 min$^{-1}$ (the latter chosen so as to keep a similar maximum:basal rate ratio as that of $r_{\text{Rep\_to\_Int}}$):

$$r_{\text{Int\_to\_Rep}} = 4.11P_R + 0.67(1 - P_R) \tag{2}$$

We assume that repressors disrupt the assembly of transcription machinery[62] to such a degree that conversion from Intermediate to Active does not occur if even a single repressor is bound. In the absence of repressors, activators facilitate the assembly of transcription machinery[63]. Brown et al.[57] reported that the rate of transcription machinery assembly is 3.3 min$^{-1}$ for a constitutively active *PHO5* promoter, and 0.025 min$^{-1}$ when the *PHO4* activator of the *PHO5* promoter is knocked out. We use this range to set

$$r_{\text{Int\_to\_Act}} = 3.3P_{A\_no\_R} + 0.025P_{\text{notA\_no\_R}}, \tag{3}$$

where $P_{A\_no\_R}$ is the probability of having no repressors and either one (for an AND-gate-incapable gene) or two (for an AND-gate-capable gene) activators bound, and $P_{\text{notA\_no\_R}}$ is the probability of having no TFs bound (for AND-gate-incapable genes) or having no repressors and not more than one activator bound (for AND-gate-capable genes).

The promoter sequence not only determines which specific TFBSs are present, but also influences nonspecific components of the transcriptional machinery[64,65]. We capture this via gene-specific but TF-binding-independent rates $r_{\text{Act\_to\_Int}}$ with which the machinery disassembles and a burst of transcription ends. In other words, we let TF binding regulate the frequency of bursts of transcription, while other properties of the *cis*-regulatory region regulate their duration. For example, the yeast transcription factor *PHO4* regulates the frequency but not duration of bursts of *PHO5* expression, by regulating the rates of nucleosome removal and of transition to but not from a transcriptionally active state[57]. Parameterization of $r_{\text{Act\_to\_Int}}$ is described in the Supplementary Methods.

**mRNA and protein dynamics.** All genes in the Active state initiate new transcripts stochastically at rate $r_{\text{max\_transc\_init}} = 6.75$ mRNA per min[57], while the time for completing transcription depends on gene length (see Supplementary Methods for parameterization of gene length and associated delay times). We model a second delay before a newly completed transcript produces the first protein, which we

assume is dominated by translation initiation (length-independent) plus elongation (length-dependent) and not splicing or mRNA export (see Supplementary Methods). After the second delay, we model protein production as continuous at a gene-specific rate $r_{\text{protein\_syn}}$ (see Supplementary Methods).

Protein transport into the nucleus is rapid[66] and is approximated as instantaneous and complete, so that the newly produced protein molecules immediately increase the probability of TF binding. Each gene has its own mRNA and protein decay rates, initialized from distributions taken from data (see Supplementary Methods).

All the rates regarding transcription and translation are listed in Supplementary Table 1, including distributions estimated from data, and hard bounds imposed to prevent unrealistic values arising during evolutionary simulations.

**Gene expression simulation.** Our algorithm is part stochastic, part deterministic. We use a Gillespie algorithm to simulate stochastic transitions between Repressed, Intermediate, and Active chromatin states, and to simulate transcription initiation and mRNA decay events. Fixed (i.e. deterministic) delay times are simulated between transcription initiation and completion, and between transcript completion and the production of the first protein. Protein production and degradation are described deterministically with ODEs. We update protein production rates frequently in order to recalculate TF concentrations and hence chromatin transition rates, limiting the magnitude of errors in the simulation (Supplementary Fig. 12). Details of our simulation algorithm are given in the Supplementary Methods. We initialize gene expression simulations with no mRNA or protein, and all genes in the Repressed state.

**Fitness.** We make fitness quantitative in terms of a "benefit" $B(t)$ as a function of the amount of effector protein $N_e(t)$ at gene expression time $t$. Our motivation is a scenario in which the effector protein is responsible for directing resources from a metabolic program favored in environment 2 to a metabolic program favored in environment 1. In environment 1, where the effector produces benefits,

$$B(t) = \begin{cases} b_{\max} \frac{N_e(t)}{N_{e\_sat}}, & N_e(t) < N_{e\_sat} \\ b_{\max}, & N_e(t) \geq N_{e\_sat} \end{cases}, \tag{4}$$

where $b_{\max}$ is the maximum benefit if all resources were redirected, and $N_{e\_sat}$ is the minimum amount of effector protein needed to achieve this. Similarly, in environment 2

$$B(t) = \begin{cases} b_{\max} - b_{\max} \frac{N_e(t)}{N_{e\_sat}}, & N_e(t) < N_{e\_sat} \\ 0, & N_e(t) \geq N_{e\_sat} \end{cases}. \tag{5}$$

We set $N_{e\_sat}$ to 10,000 molecules, which is about the average number of molecules of a metabolism-associated protein per cell in yeast[56]. Without loss of generality given that fitness is relative, we set $b_{\max} = 1$.

A second contribution to fitness comes from the cost of gene expression $C(t)$ (Fig. 1b, middle). We make this cost proportional to the total protein production rate. We estimate a fitness cost of gene expression of $2 \times 10^{-6}$ per protein molecule translated per minute, based on the cost of expressing a nontoxic protein in yeast[67] (see Supplementary Methods for details).

To ensure that gene expression changes in response to the signal, and not via an internal timer, we simulate a burn-in phase with duration drawn from an exponential distribution truncated at 30 min, with untruncated mean of 10 min. By having no fitness effects of gene expression during the burn-in, we eliminate a significant source of noise in fitness estimation due to variable burn-in duration. In our control condition, at the end of the burn-in, the signal suddenly switches to a constant ON level in environment 1, and remains off in environment 2.

We simulate gene expression for 90 min plus the duration of the burn-in (Fig. 3). A "cellular fitness" in a given environment is calculated as the average instantaneous fitness $B(t) - C(t)$ over the 90 min. We consider environment 2 to be twice as common as environment 1 (a signal should be for an uncommon event rather than the default), and take the corresponding weighted average.

**Evolutionary simulation.** We simulate a novel version of origin-fixation (weak-mutation-strong-selection) evolutionary dynamics, i.e. the population contains only one resident genotype at any time, and mutant genotypes are either rejected or chosen to be the next resident (Fig. 1c). Despite the fact that our mutant acceptance rule (see below) was chosen to maximize computational efficiency, our model usually takes 10 CPUs 1–3 days to complete an evolutionary simulation. We note that genetic homogeneity entails ignoring some important population genetic phenomena. First, if there were recombination, heterogeneity would favor mutations that combine well with a range of other genotypes. Second, clonal interference would shift evolution toward beneficial mutations of larger effect[68] (an effect we can mimic by modifying the value $10^{-8}$ in Eq. 6). Third, polymorphic populations would evolve mutational robustness[69]. None of these three effects seems a priori likely to change our conclusions, although the possibility cannot be ruled out.

Estimators $\tilde{F}$ of genotype fitness are averages of the cellular fitness values of 200 replicate simulations of gene expression per environment in the case of the mutant, plus an additional 800 should it be chosen to be the next resident. The mutant

replaces the resident if

$$\frac{\hat{F}_{\mathrm{mutant}} - \hat{F}_{\mathrm{resident}}}{|\hat{F}_{\mathrm{resident}}|} \geq 10^{-8}. \qquad (6)$$

This differs from Kimura's[70] equation for fixation probability, but captures the flavor of genetic drift. Genetic drift allows slightly deleterious mutations to occasionally fix, and beneficial mutations to sometimes fail to do so, even as the probability of fixation is monotonic with fitness. This is also achieved by our procedure, because of stochastic deviations of $\hat{F}$ from true genotype fitness. The number of gene expression simulation replicates captures the flavor of effective population size.

Note that it is possible, especially at the beginning of an evolutionary simulation, for relative fitness to be paradoxically negative. This occurs when a randomly initialized genotype does not express the effector (garnering no fitness benefit), but does express other genes (accruing a cost of expression); this combination makes fitness negative. In this rare case, for simplicity, we use the absolute value of $\hat{F}$ on the denominator.

Evolutionary simulations would require much more computation if we used a classic Wright–Fisher or Moran individual-based model, e.g. of population size 1000. Our scheme ensures that all mutations are evaluated by at least 200 gene expression simulations, making the probability of fixation of a beneficial mutation much higher than the $O(s)$ in an individual-based model. An individual-based model would also require more than the 1000 gene expression simulations required by our scheme per successful selective sweep.

If 2000 successive mutants are all rejected, the simulation is terminated; upon inspection, we found that these resident genotypes had evolved to not express the effector in either environment. We refer to each change in resident genotype as an evolutionary step. We stop the simulation after 50,000 evolutionary steps; at this time, most replicate simulations seem to have reached a fitness plateau (Supplementary Fig. 13); we analyze all replicates except those terminated early. To reduce the frequency of early termination in the case where the signal was not allowed to directly regulate the effector, we used a burn-in phase selecting on a more accessible intermediate phenotype (see Supplementary Methods). In this case, burn-in occurred for 1000 evolutionary steps, followed by the usual 50,000 evolutionary steps with selection for the phenotype of interest (Supplementary Fig. 13, right panels). Most replicates found a stable fitness plateau within 10,000 evolutionary steps, although some replicates were temporarily trapped at a low-fitness plateau (Supplementary Fig. 13).

**Genotype initialization**. We initialize genotypes with three activator genes, three repressor genes, and one effector gene. Cis-regulatory sequences and consensus binding sequences contain As, Cs, Gs, and Ts sampled with equal probability. Rate constants associated with the expression of each gene are sampled from the distributions summarized in Supplementary Table 1.

**Mutation**. A genotype is subjected to five broad classes of mutation, at rates summarized in Supplementary Table 2 and justified in the Supplementary Methods. First are single nucleotide substitutions in the cis-regulatory sequence; the resident nucleotide mutates into one of the other three types of nucleotides with equal probability. Second are single nucleotide changes to the consensus binding sequence of a TF (including the signal), with the resident nucleotide mutated into recognizing one of the other three types with equal probability. Both of these types of mutation can affect the number and strength of TFBSs.

Third are gene duplications or deletions. Because computational cost scales steeply (and nonlinearly) with network size, we do not allow effector genes to duplicate once there are five copies, nor TF genes to duplicate once the total number of TF gene copies is 19. We also do not allow the signal, the last effector gene, nor the last TF gene to be deleted.

Fourth are mutations to gene-specific expression parameters. Most of these ($L$, $r_{\mathrm{Act\_to\_Int}}$, $r_{\mathrm{protein\_syn}}$, $r_{\mathrm{mRNA\_deg}}$, and $r_{\mathrm{protein\_deg}}$) apply to both TFs and effector genes, while mutations to the gene-specific values of $K_{\mathrm{d}}(0)$ apply only to TFs and the signal. Each mutation to $L$ increases or decreases it by 1 codon, with equal probability unless $L$ is at the upper or lower bound. Effect sizes of mutations to the other five parameters are modeled in such a way that mutation would maintain specified log-normal stationary distributions for these values, in the absence of selection or arbitrary bounds (see Supplementary Methods for details). Upper and lower bounds (Supplementary Methods) are used to ensure that selection never drives these parameters to unrealistic values.

Fifth is conversion of a TF from being an activator to being a repressor, and vice versa. The signal is always an activator, and never converts.

Importantly, this scheme allows for divergence following gene duplication. When duplicates differ due only to mutations of class 4, i.e. protein function is unchanged, we refer to them as "copies" of the same gene, encoding "protein variants". Mutations in classes 2 and 5 can create a new protein. When scoring network motifs, we require two nodes to be different genes, rather than copies of the same gene (see Supplementary Methods for details).

Supplementary Table 6 summarizes the tendencies of different mutation types to be accepted, and to contribute to evolution. Acceptance rates are high, indicative of substantial nearly neutral evolution, in which slightly deleterious mutations are fixed and subsequently compensated for.

**Reporting summary**. Further information on research design is available in the Nature Research Reporting Summary linked to this article.

## Data availability
Data that can be used to recreate the presented figures are available at https://github.com/MaselLab/network-evolution-simulator.

## Code availability
Source code in C is freely available at https://github.com/MaselLab/network-evolution-simulator.

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

## Acknowledgements

Work was supported by the University of Arizona and by a Pew Scholarship to J.M., John Templeton Foundation grant 39667 to J.M., and by National Institutes of Health grants R35GM118170 to M.L.S. and R01GM076041 to J.M. We thank Hinrich Boeger for helpful discussions and careful reading of the manuscript, Jasmin Uribe for early work on this project, and the high-performance computing center at the University of Arizona for generous allocations.

## Author contributions

K.X. and J.M. designed the simulations, analyzed the results, and wrote the manuscript. K.X. performed the simulations and statistical analyses. K.X., A.K.L., and J.M. wrote the simulation code. M.L.S. and J.M. conceptualized the initial design of the simulations.

## Additional information

**Competing interests:** The authors declare no competing interests.

