## [Peer Review File · Nature Communications]

Reviewers' comments:

Reviewer #1 (Remarks to the Author):

This work presents a model and simulations of evolution of simple modules. The main goal of the paper is to show how modules such as Feed forward loop can realistically evolve under the control of natural selection. The manuscript introduces a model, then proceeds with evolution arguing that, indeed, such "AND-gated C1-FFLs" module evolve (significantly more than in a control simulation) in response to a selective pressure favouring response to long stimuli but not to short ones. The model then examines what happens when the signal is not allowed to bind to the cis-regulatory sequence of the effector gene. 4-gene modules then evolve, and it is argued that "AND-gated isolated diamonds" are better since they are not sensitive to spurious noise internal to the network.

Main concerns:

* Overall, I find the results rather unsurprising and I am not convinced such a complicated formalism is needed to get them. The fact that "AND-gated isolated diamond" are good is not a surprise at all, this is just another way to implement the idea that coherent feed forward loop with different speed and one AND gate would filter out spurious signals. In some sense the title itself is rather underwhelming, I would say diamond are a flavour of feedforward loops, where you control speed in each branch in different ways. To some extent the most "surprising" result of the paper is that they are better than 4-gene modules with more internal connections, but this probably does not require evolutionary simulations to figure out.

* I understand the desire of the authors to build a "realistic" model of evolution, however the authors do not really get any strong insights from the details of the model, since all conclusions are based on "topological" features of the network. Perhaps we don't need such intricate machinery to find only the type of motifs that arise.

* I can see issues with the assumptions of the model. For instance, since the authors require two activators to be present for the effector genes to be active (why?), it is de facto much easier to evolve an AND gate than, say, an OR gate (which requires 4 binding sites as can be seen on Fig 3B). I am not convinced this is realistic biologically and I think this assumption should be motivated to convince me the model is realistic. With this, de facto evolution is biased towards AND gate and I am not surprised at all the correct function evolves. In reality, I would expect multiple binding sites to change cooperativity in the regulation but not so easily transcriptional logic.

More minor comments

* Several times, it is mentioned that motifs are "scored" but it is never really told how, in particular it is not clear why it is sometimes more difficult.

* I find the paper very jargonistic. "C1-FFL", "AND-gated isolated diamond" seem to me too complicated terms. I would recommend listing all modules, giving them a more generic or more identifiable name for the sake of discussions. Diamonds should be defined when first introduced.

* The model overview is almost unreadable as is. It is both way too detailed and does not make a good job in pointing out the most important features of the model. I would recommend to put most of it in Supplement and to explain better the most relevant features of the model for the future discussion

* How heavy are the simulations? Is the model easily scalable for more complex problems?

* There has actually been at least one previous attempt to evolve networks performing the function described in the paper, check Pareto evolution of gene networks: an algorithm to optimize multiple fitness objectives, A Warmflash, P Francois, ED Siggia, Physical biology 9 (5), 056001, 2012

Comments on Figure

* I think Fig 1 is mostly useless since only the first network topology is ever discussed in the paper. I think it would be better to list somewhere all network topologies that will be discussed,

with their chosen name so that the reader can easily know what names such as "AND-gated isolated diamond" refer to.

* What is the "blue cloud" on top of Figure 2 ? I found the expression "developmental time" confusing, there is no real development in the biological sense. In Fig 4 there is a time axis, is it the same time ?

* Some illustration on how the evolution itself is implemented (mutations, etc...) could be of interest (e.g. in Fig 2).

Reviewer #2 (Remarks to the Author):

Xiong et al. have done an admirable job at re-igniting the FFL motif evolution debate. They tackle the question what network motifs may arise under selection if one sets the fitness function of filtering out spurious signals. They designed a model with detailed mechanics of the regulatory dynamics, and to a lesser degree for transcription, translation, and gene product degradation. I especially appreciate this part of the model, it is a rigorous approach that integrates a lot of physical knowledge of how we think transcriptional regulation works.

The manuscript is well-written and rather easy to understand (for someone with a background in evolutionary simulations). I do not think it finalizes the debate on the origin of FFLs, but I do think it is an interesting result that should be published. I have some concerns, of course, that I elaborate on below.

MAJOR ISSUES

INTRODUCTION

- I miss a section in the Introduction or in the Discussion of how the functioning of network motifs critically depends on the network's parameterization. Regarding existing literature, I would have expected references to (among others) Wall 2005, J Mol Biol 349; Wall 2011, Biosci 231; and Ingram 2006, BMC Genomic. As in the current study the authors start with the function and get the topology as a result, these works could be used to strengthen the manuscript.

METHODS

L325. I'm concerned about the environments. They seem to always have the signal switch at the same time in the graphs shown ($t=10\text{min}$). Such a constant timing can be easily exploited by evolution and give you networks with some built-in timer. Or am I misunderstanding what the environments are? Maybe you mean that environment 1 = signal on, env 2 = signal off, and you stochastically switch between the two? If so, what are the rates? 0.33 for env 1 and 0.67 for env 2, right? In other words, the section "selection conditions" needs clarification.

L372. Why do you take such a population genetics approach to a simulation study? If I play the devil's advocate, I would say you are not doing evolutionary simulations at all, but a hill climbing procedure. Sure, in popgen the weak-mutation-strong-selection scenario may be accepted, but they do that to make the maths analytically tractable. A simulation study does not have that limitation -- in fact, it is one of the strengths of simulation studies to not have that limitation. As a consequence, the current scheme lacks clonal interference/polymorphisms and does not allow for deleterious mutations (also genetic drift is missing). Any secondary effects, such as mutational robustness, are not possible in the current scheme. I have difficulties accepting the approach as a "simulation of evolution", I would rather say that the authors show mutational paths exist for FFLs.

L386. The "paradoxical negative relative fitness" needs a short explanation.

RESULTS & DISCUSSION

I have a bunch of general questions/comments:

- how biased is the outcome of "and-gate" FFLs given the starting conditions? Given that the effector is required to have two activating TFBSs in order to do something, and the AND-gate solution seems to require just that, it appears to be the first solution available...
- how robust are the final networks to parameter changes? Did you do any a posteriori sensitivity analysis? Or some sampling of networks with slightly different parameter values. I know you have tested some networks for an additional gene (Supp mat 11 "Perturbing network motifs"). But what about the other parameters that can evolve?
- evolutionary simulations of networks tend to make big networks with many genes that are hard to understand. Why do I not see these here?
- I would love to see a bunch of fitness trajectories over time. For instance, to see how often there are neutral periods where the network "meanders" across the fitness landscape without gaining fitness.
- do the high-fitness networks evolve from weak-TFBSs networks? Or a more general question: are there few or many mutational trajectories that lead to a high-fitness network? Some insight into the mutational paths that lead to FFLs of a given logic would strengthen the story.
- and how often are different types of mutations accepted? The rates listed in table 2 indicate how often mutations happen, but how often they are fixed can be quite different. That would give additional insight in the mutational paths and the structure of the fitness landscape.
- is it possible for more than one type of network logic to be present, and depending on inherent stochastics and weak binding sites, one or the other is chosen/detected? A single line in the caption of Fig 5 seems to say "yes". Or does it mean that networks neutrally evolve from one logic to another? (since you integrate over the last 10k accepted mutations)
- finally, do the authors think that with the knowledge that we gained from this detailed model, a simpler model could be derived that takes only the key elements of why FFLs may evolve? If yes, what would be the main components of that simple model? This would be an interesting discussion point.

L486. What happens to non-AND-gated FFLs that undergo the perturbation treatment? Do they maintain their function, or are they destroyed as well?

L503. The case of "neutrality" is better described as "mutations only". In my experience, when one talks about neutrality, it means the mutation does not change the selection coefficient, not that selection is absent. To avoid confusion, I would rename the term.

MINOR ISSUES

- Figure 3A. In the case of overlapping TFBSs, a translucent yellow box with a dashed outline (or something similar) that shows the overlap would be helpful. Perhaps even add a red cross to signal that it is forbidden?
- Figure 3B The meaning of the colours red and yellow changes from 3A. I would change the colours in 3A to separate them from the rest of the manuscript, where you use red for the signal etc.
- Figure 3B. Legend for the arrow styles is missing.
- Figure 5B,C; Fig 7, Fig 8. "Probability of presence" graphs would benefit from an actual number of occurrences in the graph itself, not just in the caption.

- Figure 6. Colour coding in A and B has nothing to do with each other, but (on my printed version) they look very similar.

- Some figures report "frequency of outcomes" while they appear to be histograms  "number of evolutionary outcomes". See Fig 5A, Fig S3A,B, S4, S5, S6, S8.

- The same figures Figure 5A, S3A,B, S4, etc. The x-axis does not start at zero, while it does start at zero for other graphs and the y-axis does start at zero (so there's a zero close to the origin). It would be very helpful, if the authors introduce some symbol that shows the x-axis is "broken". A squiggle, spark or double diagonal bar perhaps?

L162. Please quickly mention what M means as a unit (molar conc, mol/liter). I don't think every reader is immediately familiar with it (I was not)

L173. $K_d(1)$ and $K_d(2)$ can use a bit of elaboration, fi. "the dissociation constants with 1 and 2 mismatches, respectively"

L186. Please finish the calculation to aid the reader. $0.09 C_{TF} \sim 10\%$

L190. Can you put the star on top of the d. For some time I misinterpreted it as a multiplication.

L193. Please verify the units of the constant. I think there may be a spurious M^{-1} . I did (1) K_d in mol/liter * $3e-15$ liter/nucleus = mol/nucleus, and (2) mol/nucleus * $6e23$ molecules/mol = $1.8e9$ molecules/nucleus

L194 Where did the factor 10 come from? Perhaps from L191?

L199. Going from nucleus to cell and back is a bit confusing

L202. The text is a bit out of place. Please consider including it with Transcriptional regulation (L222) as arguably that is its topic.

L323 "Selection conditions" is a weird name. Perhaps call it "fitness and selection"?

L339. Metabolic program I and II are never mentioned or introduced. I imagine, they refer to environment 1 and 2?

L445. Please repeat what "AND-gate-capable" means, as it is quite central to the work.

L446. "motifs" is a bit of a vague term. Perhaps say "binding motifs in the upstream region of the effector gene"? Or do you want to refer to network motifs?

L452. First "both" is one too many.

- In Figure S1 I would be curious to see not only the average, but also the spread (sdev) of expression. Also, have you checked that the distribution of expression across replicates is not bimodal (i.e. behaves "normally")?

- In Supp Mat, to calculate the binding pattern, I was wondering in which order does recursion work? Do you simply go from 5' to 3' and add TFs as you encounter binding sites? The order of evaluating which binding site to add can impact the regulation.

Reviewer #1 (Remarks to the Author):

This work presents a model and simulations of evolution of simple modules. The main goal of the paper is to show how modules such as Feed forward loop can realistically evolve under the control of natural selection. The manuscript introduces a model, then proceeds with evolution arguing that, indeed, such “AND-gated C1-FFLs” module evolve (significantly more than in a control simulation) in response to a selective pressure favouring response to long stimuli but not to short ones. The model then examines what happens when the signal is not allowed to bind to the cis-regulatory sequence of the effector gene. 4-gene modules then evolve, and it is argued that “AND-gated isolated diamonds” are better since they are not sensitive to spurious noise internal to the network.

Main concerns:

*** Overall, I find the results rather unsurprising and I am not convinced such a complicated formalism is needed to get them. The fact that “AND-gated isolated diamond” are good is not a surprise at all, this is just another way to implement the idea that coherent feed forward loop with different speed and one AND gate would filter out spurious signals. In some sense the title itself is rather underwhelming, I would say diamond are a flavour of feedforward loops, where you control speed in each branch in different ways.**

We agree the diamonds are a flavor of feedforward loops, and have accordingly changed the title to *“Feed-forward regulation adaptively evolves via dynamics rather than topology when there is intrinsic noise”*, as well as referring specifically to 3-node vs 4-node motifs in the abstract, and minor changes in the Introduction. Note that common practice in the study of motifs is to use network topology for their identification, in a process that explicitly excludes diamonds when the source of the signal is not recognized as being a node. In other words, current practice implicitly indicates that our results are not anticipated.

To some extent the most “surprising” result of the paper is that they are better than 4-gene modules with more internal connections, but this probably does not require evolutionary simulations to figure out.

Our paper is NOT about the claim that the diamond is “better” than alternative topologies (which we agree would not require evolutionary simulations), but rather delineates when diamonds and C1-FFLs are and are not likely to evolve (which does require evolutionary simulations). These are not the same type of claim, with the latter requiring the admittedly complicated methodology we have developed for this paper.

As a thematic response, we suspect that Reviewer 1’s words here indicate a belief that the only question that matters is whether A is better than B, i.e. that the question of whether A evolves more readily than B is not of interest in itself (or perhaps as just one line of evidence that A might be better than B). We strongly disagree. Just because A is better than B, it doesn’t mean that A will evolve more often than B, and the question of what does and doesn’t evolve is of intense interest in biology.

We have provided extensive citations of evolutionary biologists and others making these points, and stating that they were for this reason unconvinced by current evidence on network function (Wagner 2003; Artzy-Randrup et al. 2004; Mazurie et al. 2005; Kuo et al. 2006; Solé & Valverde 2006; Lynch 2007; Knabe et al. 2008; Jenkins & Stekel 2010; Tsuda & Kawata 2010; Ruths & Nakhleh 2013; Payne & Wagner 2015). Like Reviewer 2, we do not believe the debate is over until these critiques have been addressed.

The fact that some scientists do not require this evidence, and are thus unsurprised when it finally comes in as confirmatory, does not mean that confirmatory evidence was not required.

*** I understand the desire of the authors to build a “realistic” model of evolution, however the authors do not really get any strong insights from the details of the model, since all conclusions are based on “topological” features of the network. Perhaps we don't need such intricate machinery to find only the type of motifs that arise.**

As a second thematic response, we note that while Reviewer 1 sees the complicated nature of the formalism as a weakness, Reviewer 2 sees it as a strength. We see it as an unfortunate necessity for capturing how mutational biases, rather than selection alone, shape network topology and dynamics, as well as for capturing the “internal” selection pressures imposed by intrinsic noise. Capturing mutational biases is key for thematic response #1 above. For the last 15 years, evolutionary biologists have levelled the same critique at the field, and have not received an answer. We were able to address this critique only by building this intricate machinery.

More specifically, it is simply not correct that all our conclusions are based on topological features of the network. In particular, the analysis of diamonds relies on using their evolved parameter values to distinguish between the fast path and the slow path.

It is also possible to use the model to ask a much broader range of non-topological questions. However, the paper is already extremely long. Part of its contribution is a focused answer to a long-standing question about the adaptive vs. non-adaptive evolution of feed-forward motifs. The other part is a methodological contribution which we hope will enable future work (including at least one other paper in the works by us).

*** I can see issues with the assumptions of the model. For instance, since the authors require two activators to be present for the effector genes to be active (why?),**

This requirement is part of the definition of AND-regulation, and AND-regulation is part of proposed C1-FFL function, making this requirement necessary.

it is de facto much much easier to evolve an AND gate than, say, an OR gate (which requires 4 binding sites as can be seen on Fig 3B).

This tendency is weaker than the reviewer suggests. It is not difficult to add TFBSs in a model in which the number expected for a given TF in a cis-regulatory region is approximately one. From a Poisson distribution, the presence by chance alone of two TFBSs is only about half that of one TFBS. This is an example of where our complicated mechanistic modeling details are able to give insight into pathways of evolution, and what is and is not much easier to evolve.

I am not convinced this is realistic biologically and I think this assumption should be motivated to convince me the model is realistic. With this, de facto evolution is biased towards AND gate and I am not surprised at all the correct function evolves.

The way we find out what “de facto evolution is biased towards” given our assumptions is not by verbal reasoning but by conducting rigorous control experiments to account for any mutational bias, e.g. towards AND-gates. Notably in Figure 6, C1-FFLs evolved under neutral conditions are mostly non-AND-

gated. Similarly, we get fewer AND-gated C1-FFLs in low fitness replicates than in high fitness replicates in **Fig. 5**.

This ties in to the thematic response #1 that evolutionary simulations, especially their control conditions, are necessary in reaching conclusions. No amount of biological motivation regarding the realism of model assumptions would ever be sufficient to rule out bias introduced by the specifics of those assumptions. What is sufficient is well-designed controls, where the same bias is present in both control and test conditions. And intuition about what those biases are can be wrong, as the contrast between the reviewer's intuitions and the results of our control experiments makes clear.

In reality, I would expect multiple binding sites to change cooperativity in the regulation but not so easily transcriptional logic.

We do not claim that our model assumptions are perfectly realistic. Instead they are a compromise between the minimum functionality required and avoiding unnecessary complication. We used the simplest possible implementation of logic gates we could devise. Our model choice can be interpreted as cooperativity at the level of recruitment of transcriptional machinery, whereby the effect of one bound activating TF is negligible but that of two is substantial. Modeling cooperativity in the biophysics of TF binding to TFBSs would be possible, but significantly more complicated to implement.

More minor comments

*** Several times, it is mentioned that motifs are “scored” but it is never really told how, in particular it is not clear why it is sometimes more difficult.**

We now correctly point to Supplementary Materials Section 11 for this detail. The previous submission incorrectly pointed to Methods.

We suspect the reviewer was confused by the section title “*Weak TFBSs make motif scoring more difficult*” in the main text of the last submission, expecting to find the information there. Scoring motif presence/absence from simulation outcomes is quite straightforward and not particularly interesting, hence the relegation of this portion of the Methods to the Supplement. What is more subtle and difficult, and hence required discussion in the main text, is to differentiate which network motifs are adaptive/functional, given that a weak TFBS might be present by chance and alter topological scoring while having little impact on function. E.g., what we term a near-AND-gated network motif is one that would be an AND-gated motif if it were not for the presence of a weak TFBS; this binding site might be inconsequential enough such that the motif functions much like an AND-gated motif.

We have clarified the section title to read “*Weak TFBSs can change how adaptive motifs are scored even when they do not change function*”. We also removed confusing language from the abstract.

*** I find the paper very jargonic. “C1-FFL”, “AND-gated isolated diamond” seem to me too complicated terms. I would recommend listing all modules, giving them a more generic or more identifiable name for the sake of discussions.**

We are not sure what alternative names to use for the motifs. Obviously we need terms to describe each of the primary objects we study. C1-FFL is standard in the field, and changing that to something else would not result in our using any fewer specialized terms, but simply make one of those terms less comprehensible to those who have read other papers. “Diamond” is less standard because there has

been much less previous work on this motif, but we still chose the term based on some precedent, specifically Burda et al. 2011 and Alon 2007a. An alternative name for the motif is a 4-node FFL, as used by Ma'anya et al. 2005, but we believe that contrasting two motifs with similar names that both include "FFL" would make the paper more difficult rather than less difficult to read, i.e. that the more pronounced contrast between FFL and diamond makes meaning easier to parse. We are open to alternatives to our new term "isolated diamond" to describe topologies that contain only a diamond and not also a C1-FFL, but we do require some term to describe that more specific topology, and there appears to be no precedent.

Diamonds should be defined when first introduced.

It is not possible to define diamonds within the abstract, which is not a traditional home for rigorous definitions, but an informal flavor of its meaning is now given. We have removed reference to diamonds in the Introduction, so that its definition is no longer called for there. The next use is a subtitle, and we define diamond shortly thereafter:

"While visually examining the network context of these C1-FFLs, we discovered that many were embedded within AND-gated "diamonds". In a diamond, the signal activates the expression of two genes that encode different TFs, and the two TFs activate the expression of an effector gene (Fig. 7A middle)."

*** The model overview is almost unreadable as is. It is both way too detailed and does not make a good job in pointing out the most important features of the model. I would recommend to put most of it in Supplement and to explain better the most relevant features of the model for the future discussion**

We have moved a number of components of the model overview (i.e. the main text Methods section) to the Supplement. These include the tables of parameter value estimates (a move in any case necessary to meet the *Nature Communications* limit of 70 main text references), and details of TF binding and of delay times in transcription and translation. Shorter summaries of the moved material are now included in the revised model overview.

*** How heavy are the simulations? Is the model easily scalable for more complex problems ?**

We note in the Discussion:

"Our model, while powerful in some ways, is computationally limited to small TRNs."

We also added a sentence to Model Overview to describe the computational cost:

"Our model usually takes 10 CPUs 1-3 days to complete an evolutionary simulation."

*** There has actually been at least one previous attempt to evolve networks performing the function described in the paper, check Pareto evolution of gene networks: an algorithm to optimize multiple fitness objectives, A Warmflash, P Francois, ED Siggia, Physical biology 9 (5), 056001, 2012**

We now discuss this paper in the Discussion:

“Previous work has also identified alternatives to AND-gated C1-FFLs. Specifically, in mixed networks of transcriptional regulation and protein-protein interactions, FFLs did not evolve under selection for delayed turn-on (as well as rapid turn-off)⁵⁷. Indeed, even when a FFL topology was enforced, with only the parameters allowed to evolve, two alternative motifs remained superior⁵⁷. However, one alternative motif, which the authors called “positive feedback” is essentially still an AND-gated C1-FFL, specifically one in which the intermediate TF expression is also AND-gated, requiring both itself and the signal for upregulation. The other is a cascade in which the signal inhibits the expression of an intermediate TF protein that represses the expression of the effector. The cost of constitutive expression of the intermediate TF in the absence of the signal was not modeled⁵⁷, giving this cascade an unrealistic advantage.”

Comments on Figure

*** I think Fig 1 is mostly useless since only the first network topology is ever discussed in the paper. I think it would be better to list somewhere all network topologies that will be discussed, with their chosen name so that the reader can easily know what names such as "AND-gated isolated diamond" refer to.**

We have moved **Fig. 1**, whose purpose is to illustrate the meaning of “C1” in “C1-FFL”, to the Supplement, also making improvements to it so it better fulfils that purpose. What is now Figure 2 requires this figure (and not just the first topology in it) in order for all parts of it to be comprehensible, so its legend now refers to **Fig. S1** for what used to be **Fig. 1**.

The problem with comparing C1-FFLs and diamonds from the outset is that it would require distinguishing between direct and indirect regulation, which does not make sense so early in the paper. Instead, we have removed discussion of diamonds from the Introduction, so that they are not referred to before being explained. The Figure that the reviewer is looking for is what is now **Fig. 7a**, which comes at an appropriate stage of the narrative, immediately after all of its components and the methods to study them have been introduced.

*** What is the “blue cloud” on top of Figure 2 ?**

It is the pre-initiation complex of transcription. We now clearly label it in what is now **Fig. 1**.

I found the expression “developmental time” confusing, there is no real development in the biological sense.

“Development” means different things in different parts of biology – sometimes, especially in an abstract model, it is any process of going from genotype to phenotype given environmental influences, while in other fields the term is a restricted subset of this, e.g. restricted to multicellular morphologies. We now more clearly contrast “developmental time” with “evolutionary time” at its first appearance, which is the purpose of the term in our context. We are not aware of another suitable term to contrast with “evolutionary time”. Just referring to “time” in our model would be ambiguous.

In Fig 4 there is a time axis, is it the same time ?

Yes. We have changed the x axis label of what is now **Fig. 3** to “Developmental time (min)”. The figure is also refined as per reviewer 2’s comment.

*** Some illustration on how the evolution itself is implemented (mutations, etc...) could be of interest (e.g. in Fig 2).**

We have added an illustration of the evolutionary level of the simulation as **Fig. 1c**.

Reviewer #2 (Remarks to the Author):

Xiong et al. have done an admirable job at re-igniting the FFL motif evolution debate. They tackle the question what network motifs may arise under selection if one sets the fitness function of filtering out spurious signals. They designed a model with detailed mechanics of the regulatory dynamics, and to a lesser degree for transcription, translation, and gene product degradation. I especially appreciate this part of the model, it is a rigorous approach that integrates a lot of physical knowledge of how we think transcriptional regulation works.

The manuscript is well-written and rather easy to understand (for someone with a background in evolutionary simulations). I do not think it finalizes the debate on the origin of FFLs, but I do think it is an interesting result that should be published. I have some concerns, of course, that I elaborate on below.

MAJOR ISSUES

INTRODUCTION

- I miss a section in the Introduction or in the Discussion of how the functioning of network motifs critically depends on the network's parameterization.

Our new results on evolutionary constraint on parameter values (see response below to comment on robustness to parameter value changes) add substance in this regard, albeit not in the Introduction or Discussion.

Regarding existing literature, I would have expected references to (among others) Wall 2005, J Mol Biol 349; Wall 2011, Biosci 231; and Ingram 2006, BMC Genomic. As in the current study the authors start with the function and get the topology as a result, these works could be used to strengthen the manuscript.

We have cited these articles as part of a significantly rearranged Discussion:

"The function of a motif relies ultimately on its dynamic behavior, with topology merely a means to that end. To create two pathways that regulate the effector in different speeds, the C1-FFL motif uses a pair of short and long pathways, but these also correspond to fast-degrading and slow-degrading TFs. This same function was achieved entirely non-topologically in our adaptively evolved diamond motifs. This agrees with other studies showing that topology alone is not enough to infer activities such as spurious signal filtering from network motifs⁶⁸⁻⁷⁰."

METHODS

L325. I'm concerned about the environments. They seem to always have the signal switch at the

same time in the graphs shown (t=10min). Such a constant timing can be easily exploited by evolution and give you networks with some built-in timer.

The reviewer's concern is legitimate; we did have constant timer that always switched at exactly t=10min, which does risk the evolution of networks with a built-in timer rather than the evolution of a response to the signal. To remove this risk, when selecting for filtering out a spurious signal, we no longer initialize with the signal off. Instead, the signal remains off for a period of time drawn from an exponential distribution truncated at 30 minutes with an untruncated mean of 10 minutes. This is followed by a short spurious on signal that is, as before, of exactly 10 minutes duration.

We have re-run all simulations under this new setup.

Or am I misunderstanding what the environments are? Maybe you mean that environment 1 = signal on, env 2 = signal off, and you stochastically switch between the two? If so, what are the rates? 0.33 for env 1 and 0.67 for env 2, right? In other words, the section "selection conditions" needs clarification.

No, you understood correctly. We now clarify this in text:

*"Fitness is a weighted average across **separate developmental simulations** in these two environments."*

The numbers 0.33 and 0.67 refer not to switching rates but to the weights used to average fitness across the two environments, each assessed independently.

L372. Why do you take such a population genetics approach to a simulation study? If I play the devil's advocate, I would say you are not doing evolutionary simulations at all, but a hill climbing procedure. Sure, in popgen the weak-mutation-strong-selection scenario may be accepted, but they do that to make the maths analytically tractable. A simulation study does not have that limitation -- in fact, it is one of the strengths of simulation studies to not have that limitation. As a consequence, the current scheme lacks clonal interference/polymorphisms and does not allow for deleterious mutations (also genetic drift is missing). Any secondary effects, such as mutational robustness, are not possible in the current scheme. I have difficulties accepting the approach as a "simulation of evolution", I would rather say that the authors show mutational paths exist for FFLs.

Agent-based simulation is far too computationally expensive to be at all accessible. Indeed, even a classic weak-mutation-strong-selection approach is *still* too computationally expensive. This is because a mutation's probability of fixation calculated from Kimura's equation is usually very low (even for a beneficial mutation), leading to too many beneficial mutations being rejected and thus wasting computation. However, we would still not characterize what we are doing as simple hill-climbing, in which all beneficial mutations and no deleterious mutations are accepted. We have added text to explain the rationale of our method:

"This differs from Kimura's ⁴⁸ equation for fixation probability, but captures the flavor of genetic drift. Genetic drift allows slightly deleterious mutations to occasionally fix, and beneficial mutations to sometimes fail to do so, even as the probability of fixation is monotonic of fitness. This is also achieved by our procedure, because of stochastic deviations of \hat{F} from true genotype fitness. The number of developmental replicates captures the flavor of effective population size."

We have added more text to acknowledge the caveats correctly identified by the reviewer:

“We note that genetic homogeneity entails ignoring some important population genetic phenomena. First, if there were recombination, heterogeneity would favor mutations that combine well with a range of other genotypes. Second, clonal interference would shift evolution toward beneficial mutations of larger effect⁴³ (an effect we can mimic by modifying the value 10^8 in the equation below). Third, polymorphic populations would evolve mutational robustness⁴⁴. None of these three effects seems a priori likely to change our conclusions, although the possibility cannot be ruled out.”

L386. The "paradoxical negative relative fitness" needs a short explanation.

We now explain in more detail how negative relative fitness comes about:

“This occurs when a randomly initialized genotype does not express the effector (garnering no fitness benefit), but does express other genes (accruing a cost of expression); this combination makes fitness negative.”

RESULTS & DISCUSSION

I have a bunch of general questions/comments:

- how biased is the outcome of "and-gate" FFLs given the starting conditions? Given that the effector is required to have two activating TFBSs in order to do something, and the AND-gate solution seems to require just that, it appears to be the first solution available...

Evolution does tend to choose “the first solution available”, so we consider making this choice to be a desirable feature reproducing a genuine biological phenomenon rather than an undesirable bias of our model. What would be undesirable is scoring a non-adaptive mutational bias as adaptive. As mentioned in the response to reviewer 1 above, we conduct control experiments (**Fig. 6**) to account for any mutational bias towards AND-gates.

- how robust are the final networks to parameter changes? Did you do any a posteriori sensitivity analysis? Or some sampling of networks with slightly different parameter values. I know you have tested some networks for an additional gene (Supp mat 11 "Perturbing network motifs"). But what about the other parameters that can evolve?

The usual method for assessing sensitivity to perturbing a parameter value of a gene can be misleading. This is because both the effector and the auxiliary TF are usually encoded by multiple copies of genes, and redundancy conceals the effect that a mutation would have in a single gene copy.

Instead of sensitivity (i.e. the effect of perturbing a parameter value), we therefore explore evolutionary constraint (i.e. the extent to which parameter values in evolved systems differ from parameter values expected from mutation alone). We quantify this as the ratio of the variance of the values of the parameter in evolved AND-gated C1-FFLs to the variance under neutral evolution. These new methods and results on constraint are described in **Tables S4-S6**, and summarized in the Results.

- evolutionary simulations of networks tend to make big networks with many genes that are hard to understand. Why do I not see these here?

Our biggest disappointment with this project was poor scaling of computation with the number of genes. For this reason, we capped the size of networks, on top of selection against unnecessary gene expression. The cap is described at the end of section Mutation, and we have now made it more clear:

“Because computational cost scales steeply (and non-linearly) with network size, we do not allow effector genes to duplicate once there are 5 copies, nor TF genes to duplicate once the total number of TF gene copies is 19.”

- I would love to see a bunch of fitness trajectories over time. For instance, to see how often there are neutral periods where the network "meanders" across the fitness landscape without gaining fitness.

Fig. S3 now shows 4 trajectories when the signal is allowed to directly regulate the effector and 4 for when it is not. We add the following to the main text:

*“Most replicates found a stable fitness plateau within 10,000 evolutionary steps, although some replicates were temporarily trapped at a low fitness plateau (**Fig. S3**).”*

- do the high-fitness networks evolve from weak-TFBSs networks? Or a more general question: are there few or many mutational trajectories that lead to a high-fitness network? Some insight into the mutational paths that lead to FFLs of a given logic would strengthen the story.

There are many analyses we could add here, but we fear the paper is already long. The reason that we exclude this in particular is that real evolution moves from solution to solution, whereas our simulations move from random networks to solution. The external validity of any findings about mutational pathways is therefore unclear. We feel that we are on more solid ground when it comes to evolutionary outcomes than mutational pathways.

- and how often are different types of mutations accepted? The rates listed in table 2 indicate how often mutations happen, but how often they are fixed can be quite different. That would give additional insight in the mutational paths and the structure of the fitness landscape.

We have added **Table S3** to summarize the frequencies with which different types of mutations are accepted at the beginning and at the end of evolutionary simulations. We describe Table 3 at the end of Mutation in the main text:

*“**Table S3** summarizes the tendencies of different mutation types to be accepted, and to contribute to evolution. Acceptance rates are high, indicative of substantial nearly neutral evolution, in which slightly deleterious mutations are fixed and subsequently compensated for.”*

- is it possible for more than one type of network logic to be present, and depending on inherent stochasticity and weak binding sites, one or the other is chosen/detected? A single line in the caption of Fig 5 seems to say "yes". Or does it mean that networks neutrally evolve from one logic to another? (since you integrate over the last 10k accepted mutations)

Multiple types of network logics do coexist, and in that case each is scored. This can for example be seen in what is now **Fig. 4B**, in which the sum of the probability of presence of the AND-gated C1-FFL and that

of the signal-controlled C1-FFL is greater than 1. The **Fig. 4** legend has been rewritten to make this point clearer.

- finally, do the authors think that with the knowledge that we gained from this detailed model, a simpler model could be derived that takes only the key elements of why FFLs may evolve? If yes, what would be the main components of that simple model? This would be an interesting discussion point.

This depends on the evolutionary questions and hypotheses at stake. For example, if one is interested in only the steady state expression levels of genes in a network (which is not the case for understanding FFLs), then stochastic gene expression may not be vital and can be stripped out. We do believe that the high prevalence of weak TFBSs and of gene duplication will be critical to all related models. We have added the following to Discussion:

“Our model, while complex for a model and hence capable of capturing intrinsic noise, is inevitably less complex than the biological reality. However, we hope to have captured key phenomena, albeit in simplified form. One key phenomenon is that TFBSs are not simply present vs. absent but can be strong or weak, i.e. the TRN is not just a directed graph, but its connections vary in strength. Our model, like that of Burda et al.⁵⁴ in the context of circadian rhythms, captures this fact by basing TF binding affinity on the number of mismatch deviations from a consensus TFBS sequence. While in reality, the strength of TF binding is determined by additional factors, such as broader nucleic context and cooperative behavior between TFs (reviewed in Inukai et al.⁵⁸), these complications are unlikely to change the basic dynamics of frequent appearance of weak TFBSs and greater mutational accessibility of strong TFBSs from weak TFBSs than de novo. Similarly, AND-gating can be quantitative rather than qualitative⁵⁹, a phenomenon that weak TFBSs in our model provide a simplified version of.”

L486. What happens to non-AND-gated FFLs that undergo the perturbation treatment? Do they maintain their function, or are they destroyed as well?

In our last submission, we had a bug in the scoring of network motifs with weak TFBSs excluded. This produced incorrectly high numbers of signal-controlled FFLs in the original **Fig. 5C** (now **Fig. 4C**). When we corrected this mistake, the occurrence of non-AND-gated FFLs fell close to zero in both high-fitness and low-fitness replicates, when weak TFBSs are excluded. This suggests that non-AND-gated FFLs are mostly functionless, removing the need to perform the requested analysis.

We apologize for the mistake, and thank the reviewer for this comment, which helped us find it and strengthen the manuscript's results!

L503. The case of "neutrality" is better described as "mutations only". In my experience, when one talks about neutrality, it means the mutation does not change the selection coefficient, not that selection is absent. To avoid confusion, I would rename the term.

We have switched to the term “no selection”, which we believe avoids all confusion, while being more consistent with the other options that describe the form of selection.

MINOR ISSUES

- **Figure 3A. In the case of overlapping TFBSs, a translucent yellow box with a dashed outline (or something similar) that shows the overlap would be helpful. Perhaps even add a red cross to signal that it is forbidden?**

This is now on its own as **Fig. S10**, and we have improved it, including removing the color that might have given the impression of vacancy vs. occupancy. The hindrance is mutual, and it was not our intention to label one as occupied and one as forbidden, but instead show them as symmetric.

- **Figure 3B The meaning of the colours red and yellow changes from 3A. I would change the colours in 3A to separate them from the rest of the manuscript, where you use red for the signal etc.**

What was **Fig. 3A** is now on its own as **Fig. S10**.

- **Figure 3B. Legend for the arrow styles is missing.**

We have added a legend to what is now **Fig. 2** to explain arrow styles.

- **Figure 5B,C; Fig 7, Fig 8. "Probability of presence" graphs would benefit from an actual number of occurrences in the graph itself, not just in the caption.**

We tried to add a number on the top of each bar, but the figure became too crowded. We think the difference between bars is important, but the exact number of occurrence of a motif should not be over-interpreted. We instead added numbers of replicates " $n=...$ " to the legend labels in **Fig. 4B,C** (formerly **Fig. 5B,C**), **Fig. 6** (formerly **Fig. 7**), and **Fig. 7** (formerly **Fig. 8**).

- **Figure 6. Colour coding in A and B has nothing to do with each other, but (on my printed version) they look very similar.**

We meant to match the color of the signal to that of the signal-controlled motif and the color of the slow TF to that of the slow TF-controlled motif. However, we set the wrong color in B. We apologize for the mistake, and have corrected the colors in what is now **Fig. 5**.

- **Some figures report "frequency of outcomes" while they appear to be histograms  "number of evolutionary outcomes". See Fig 5A, Fig S3A,B, S4, S5, S6, S8.**

We have changed "frequency" to "number".

- **The same figures Figure 5A, S3A,B, S4, etc. The x-axis does not start at zero, while it does start at zero for other graphs and the y-axis does start at zero (so there's a zero close to the origin). It would be very helpful, if the authors introduce some symbol that shows the x-axis is "broken". A squiggle, spark or double diagonal bar perhaps?**

Note that **Figs. 5A, S3, S4, S5A, S6A** are now **Figs. 4A, S4, S5, S6A, S7A**, respectively.

We have added longer ticks to the x-axes, and labeled the starting values of the x-axes in these histograms. Note that there is no special significance to zero fitness in our scheme, with negative fitness possible.

L162. Please quickly mention what M means as a unit (molar conc, mol/liter). I don't think every reader is immediately familiar with it (I was not)

Note that this now appears only in the Supplement, specifically Section 1 and Table S1. We have replaced M with mole/liter.

L173. K_d(1) and K_d(2) can use a bit of elaboration, fi. "the dissociation constants with 1 and 2 mismatches, respectively"

We have added the text suggested by the reviewer to the end of this sentence, now in the Supplement.

L186. Please finish the calculation to aid the reader. $0.09 C_{TF} \sim 10\%$

We appended " $\approx 0.1C_{TF}$ " to the equation.

L190. Can you put the star on top of the d. For some time I misinterpreted it as a multiplication.

We replaced K_d^* with \widehat{K}_d .

L193. Please verify the units of the constant. I think there may be a spurious M^{-1} . I did (1) K_d in mol/liter * $3e-15$ liter/nucleus = mol/nucleus, and (2) mol/nucleus * $6e23$ molecules/mol = $1.8e9$ molecules/nucleus

The equation gives the rescaling factor to be multiplied to K_d , so it needs a M^{-1} . We rewrote the sentence to

"We also convert \widehat{K}_d from the units of mole/liter in which K_d is estimated empirically to the more convenient molecules/nucleus. The rescaling factor r for which \widehat{K}_d (in molecule/nucleus) = $r\widehat{K}_d$ (in mole/liter) is 3×10^{-15} liter/nucleus $\times 6.02 \times 10^{23}$ molecule/mole = 1.8×10^9 molecule cell⁻¹ liter mole⁻¹. Taken together, \widehat{K}_d (molecule/nucleus) = $10rK_d$ (mole/liter), where the factor 10 accounts for non-specific TF binding."

L194 Where did the factor 10 come from? Perhaps from L191?

The factor 10 comes from converting K_d to \widehat{K}_d . See response to last comment for the revised text.

L199. Going from nucleus to cell and back is a bit confusing

We deleted the sentence. We replaced N_i in Eq. 1 with C_i in what is now Eq. S1, where C is defined as the total nucleic TF concentration in Section 1 of the Supplementary Text.

L202. The text is a bit out of place. Please consider including it with Transcriptional regulation (L222) as arguably that is its topic.

We have moved it to the beginning of Section 2 in the Supplementary Text.

L323 "Selection conditions" is a weird name. Perhaps call it "fitness and selection"?

The problem with the term “fitness” is that our values of \hat{F} are not equivalent to classic population genetics expectation of surviving progeny. For this reason we use the term cautiously in the manuscript. We now call the section simply “Selection” instead of “Selection conditions”.

L339. Metabolic program I and II are never mentioned or introduced. I imagine, they refer to environment 1 and 2?

Yes, and we have rephrased to avoid this unnecessary terminology.

L445. Please repeat what "AND-gate-capable" means, as it is quite central to the work.

It would be confusing to repeat this complex definition in the first paragraph of the Results, which necessarily has a different focus, but we do now explicitly point, in this passage, to where in the manuscript the definition can be found.

L446. "motifs" is a bit of a vague term. Perhaps say "binding motifs in the upstream region of the effector gene"? Or do you want to refer to network motifs?

Replaced “motifs” with “network motifs”.

L452. First "both" is one too many.

Deleted the first “both”.

- In Figure S1 I would be curious to see not only the average, but also the spread (sdev) of expression. Also, have you checked that the distribution of expression across replicates is not bimodal (i.e. behaves "normally")?

We thank the reviewer for this comment. The expression levels across replicates are usually approximately log-normal distributed, but the expression levels in a high-fitness genotype do transit to a bimodal distribution over time after responding to the short spurious signal. The arithmetic mean in our last submission was clearly problematic, and has been removed. Instead of replacing it with a different averaging procedure, we now directly show the expression levels in several randomly chosen replicates in what is now **Fig. S2**.

- In Supp Mat, to calculate the binding pattern, I was wondering in which order does recursion work? Do you simply go from 5' to 3' and add TFs as you encounter binding sites? The order of evaluating which binding site to add can impact the regulation.

We use a dynamical programming approach that always goes from the same end to the other end in the process of calculating the joint probability distribution, but the order does not affect the result, i.e. it would be identical if it were computed in the opposite direction. Every possible binding configuration is included in our calculation – the left to right order is merely a systematic way of proceeding through them.

REVIEWERS' COMMENTS:

Reviewer #1 (Remarks to the Author):

I am happy with the current revision and with the authors' arguments in response to my remarks in the previous round of review.

The only very minor remark I have is that I am still rather "perturbed" by the expression "developmental time", since again it means something completely different in developmental biology for instance. It is first mentioned on p 5 and it is not explained what it is. I have no obvious suggestions (maybe since equations are integrated "integration time" or even "dynamical time" could be used), or alternatively the authors could add a sentence to explicitly contrast evolutionary time with what they call "developmental time". Some other groups refer to "epoch" to tell about evolutionary time, and thus what is called here "developmental time" simply is called ... time.

Reviewer #2 (Remarks to the Author):

In this version of the manuscript, Xiong et al. have carefully addressed the concerns raised by the reviewers. The result is a manuscript with plenty of changes. The major ones are (1) a re-orientation of the paper towards dynamics over topology, (2) a thorough clarification of the Methods section (great work!), (3) improved figures.

Because of all these changes, I re-answered the default questions of Nature Communications below, followed by my comments on the new manuscript.

- What are the major claims of the paper?

The major claims are in the title: Given intrinsic noise, (1) feed-forward regulation evolves adaptively, and (2) dynamics matter more than topology.

- Are they novel and will they be of interest to others in the community and the wider field?

As far as I know, part (1) of the claims is novel and of interest to others in both the evolutionary biology and regulatory network communities. Part (2) is perhaps not novel, but in my experience too many people still claim/think topology is the determining factor, so it is good the point of dynamics over topology is driven home once again.

- If the conclusions are not original, it would be helpful if you could provide relevant references.

In my previous review, I suggested some relevant references regarding the importance of dynamics. The authors cite these in the new manuscript.

- Is the work convincing, and if not, what further evidence would be required to strengthen the conclusions?

The work is convincing and I do not think further evidence is needed for publication. Some rewriting may be needed, though.

- On a more subjective note, do you feel that the paper will influence thinking

in the field?

My hope is that the paper helps highlighting that part of understanding biological systems is to explain what does and does not evolve (as also stated by the authors in their Introduction and in their response to reviewer 1).

MAJOR CONCERNS

GENERAL REMARKS

My main concern is that this new version of the manuscript has two story lines. The old one about the evolution of an FFL topology, and the new one that wants to stress dynamics over topology. Ideally, the authors adjust their text, especially the introduction and discussion, to better reflect the new message.

ABSTRACT

- The text has not changed... I imagine the authors simply forgot to do this and will provide a new abstract in a next round of revisions.
- Also I miss a conclusion (or it is very implicit).

INTRO

- Considering the changed title -- and hence the new message, the introduction has been only minimally adjusted. The title stresses dynamics over topology, yet the intro only talks about topology. Dynamics are not even mentioned... that seems unbalanced to me.

METHODS

- I would think the work of Hermesen et al. 2006, Buchler et al. 2003, etc. are relevant to the methodology developed here. I leave it to the authors to decide if they want to cite these.

- p16, Evolutionary simulation. I disagree with the claim that modelling a heterogeneous population is out of the question. The new Fig 1 made me realise that every mutant is evaluated 200 times and every accepted mutant another 800 times. In reality, no-one is re-evaluated 200 times to establish a mean behavior. An organism is a single instance of a stochastic process. Thus a population of 200 individuals is computationally feasible, assuming that -- roughly speaking -- any tricks to optimize the re-evaluation of individuals cancel out against the 800 extra evaluations of an accepted mutant.

The key is to accept that not only evolution, but also development is a stochastic process. Surely individuals will get lucky some times and have a high fitness, while on average they would perform badly, but at some point their offspring will be unlucky and weeded out by selection.

The easiest solution is to remove the claim.

RESULTS

- I would invite the authors to consider the point of view that there may be a

so-called complexity ratchet at work in their simulations. See Liard et al. 2018 for an example (https://www.mitpressjournals.org/doi/abs/10.1162/isal_a_00051).

It would mean that the networks that have an AND-gate C1-FFL do not evolve to/from other relatively high-fitness network structures. In some sense, one would have to "get lucky" at the start of evolution and hit upon an AND-gate FFL. When a network is not lucky, it wanders into an area of the fitness landscape that favors the use of weak TFBSs and alternative logic. Somehow that complexity of many weak binding sites obstructs the process of finding AND-gates.

If (and this is a big if) the above is the case, the claim that the preference for AND gates is associated to adaptation (as is done on page 22, 1430), is still true, but not in the sense of adaptation vs mutation bias. Instead, adaptation can go into two directions, one simple and high-fitness, another complex and only sometimes high-fitness (i.e. signal controlled cases).

With respect to the three control conditions, the change in fitness function alters the shape of the landscape in terms of peaks and neutral/genotype networks (but not in terms of connectivity between individual genotypes). At first thought, the result that AND-gated FFLs are for filtering short signals, is not against the complexity ratchet. It is not specifically in favor either, so how these results fit together is a rather open question for me.

If it is easy to check what FFL logic precedes or comes after AND-gated FFLs over evolutionary time, I would be curious to know if that fits with the complexity ratchet idea. For the rest, I realize this is mostly me sharing a scientific discussion with the authors and other reviewer. There is no need for the authors to act upon this.

- p29, 1551. What mutational biases are meant?

- As remarked by the authors in their response, the manuscript is getting long. Unfortunately, I agree as I find myself struggling through the second part of the results. The issue is not that it is boring, rather I lose track of the main message of the paper and the text is a rather technical report of subtle differences between each of the network motifs and their occurrence. If the authors see a way of focussing their message and leaving out unnecessary text, that would benefit their paper.

MINOR CONCERNS

- Title: would it not be more elegant to say "Under intrinsic noise, feed-forward regulation adaptively evolves via dynamics rather than topology"?

- L55,74,478,479 etc. I do not see the need for quotes (").

- L98-100 Somehow the sentence does not read nicely, it has perhaps too much info. I would split it.

- L106-111 This sentence is loooong.

- Fig 5A, 7A. Please add the legend of Fig 2 to remind me what the arrows mean.

- 1476. I find remembering what each of the three negative controls mean

difficult. Would it be possible to add little cartoon explanations to Fig 6 that visualize how the fitness function is changed?

- Fig 9. Is there a journal policy on sub-subpanels naming? Personally, I prefer to just name them sequentially, instead of hierarchically.

SUPP INFO

- Table S3. The heading of column 3-4 is a bit double, not? An accepted mutation is given to be accepted.

- Table S3, l36. typo in "which can be easily occur"

- Table S3. Just for scientific discussion: mutations of locus length become a lot more popular in late evolution, even if they are a magnitude less frequent than gene indels. I am curious how the authors would explain that. I guess it is one of the few relatively neutral mutations on a gene's upstream region.

- Table S4. It would aid the reader if the authors explain why selection and no-selection in the ratio calculations of V and M are swapped? V_n/V_s , but M_s/M_n . I imagine because in the selection case, average parameter values tend to be greater than in the no-selection case. Variation, however, is larger in the no-selection case.

- Table S4, l46-51. The explanation of sampling could be better formulated.

- Table S4, l61. Strictly speaking also for $R_{Act_to_Int}$, $V_n/V_s < 1.0$, not just for $K_d(0)$. Taking into account Table S5 and S6, I see that is an outlier.

- Fig S12. Please check the naming of the motifs. I do not see a 'C' and 'A' is never mentioned. Also repeating the legend of Fig 2 would be a useful reminder.

- Disclaimer: I have not had the time to carefully dig through all the supplementary methods.

SOFTWARE

- I tried to compile the code with gcc, but it gets stuck at an icc specific flag, namely "-fp-model". If you claim that it should work with gcc, please fix this. Otherwise simply state that it is only guaranteed to work with icc.

With best wishes,
Anton Crombach

REVIEWERS' COMMENTS:

Reviewer #1 (Remarks to the Author):

I am happy with the current revision and with the authors' arguments in response to my remarks in the previous round of review.

The only very minor remark I have is that I am still rather "perturbed" by the expression "developmental time", since again it means something completely different in developmental biology for instance. It is first mentioned on p 5 and it is not explained what it is. I have no obvious suggestions (maybe since equations are integrated "integration time" or even "dynamical time" could be used), or alternatively the authors could add a sentence to explicitly contrast evolutionary time with what they call "developmental time". Some other groups refer to "epoch" to tell about evolutionary time, and thus what is called here "developmental time" simply is called ... time.

The paper requires disambiguation that there are two different timescales, one nested within the other, hence "time" is not a viable option. Epoch traditionally refers to macroevolutionary timescales in paleontology, which is substantially longer than the timescale at which single mutations become fixed. Not all gene expression equations are integrated, and both gene expression and evolution are dynamical, so none of these terms will work. We have replaced "developmental time" with "gene expression time" to reflect the fact that gene expression changes need not correspond to processes that a developmental biologist would recognize as development.

Reviewer #2 (Remarks to the Author):

In this version of the manuscript, Xiong et al. have carefully addressed the concerns raised by the reviewers. The result is a manuscript with plenty of changes. The major ones are (1) a re-orientation of the paper towards dynamics over topology, (2) a thorough clarification of the Methods section (great work!), (3) improved figures.

Because of all these changes, I re-answered the default questions of Nature Communications below, followed by my comments on the new manuscript.

- What are the major claims of the paper?

The major claims are in the title: Given intrinsic noise, (1) feed-forward regulation evolves

adaptively, and (2) dynamics matter more than topology.

- Are they novel and will they be of interest to others in the community and the wider field?

As far as I know, part (1) of the claims is novel and of interest to others in both the evolutionary biology and regulatory network communities. Part (2) is perhaps not novel, but in my experience too many people still claim/think topology is the determining factor, so it is good the point of dynamics over topology is driven home once again.

- If the conclusions are not original, it would be helpful if you could provide relevant references.

In my previous review, I suggested some relevant references regarding the importance of dynamics. The authors cite these in the new manuscript.

- Is the work convincing, and if not, what further evidence would be required to strengthen the conclusions?

The work is convincing and I do not think further evidence is needed for publication. Some rewriting may be needed, though.

- On a more subjective note, do you feel that the paper will influence thinking in the field?

My hope is that the paper helps highlighting that part of understanding biological systems is to explain what does and does not evolve (as also stated by the authors in their Introduction and in their response to reviewer 1).

MAJOR CONCERNS

GENERAL REMARKS

My main concern is that this new version of the manuscript has two story lines. The old one about the evolution of an FFL topology, and the new one that wants to stress dynamics over topology. Ideally, the authors adjust their text, especially the introduction and discussion, to better reflect the new message.

ABSTRACT

- The text has not changed... I imagine the authors simply forgot to do this and will provide a new abstract in a next round of revisions.

We checked the abstract, and are confused by this comment. It seemed to us to be a good balance between the two story lines. Sentence 1 is about both, sentence 2 about the first story line, sentences 3-4 about methods, sentence 5 about the first story line, sentences 6-8 about the second. We aren't sure what to change.

- Also I miss a conclusion (or it is very implicit).

As outlined above, the penultimate four sentences of the abstract all summarize main results. With only a 150 word abstract limit, it is hard to do more. We added one more subsequent sentence #9 in an attempt to integrate.

INTRO

- Considering the changed title -- and hence the new message, the introduction has been only minimally adjusted. The title stresses dynamics over topology, yet the intro only talks about topology. Dynamics are not even mentioned... that seems unbalanced to me.

We have added a new paragraph to the Introduction:

“Given the potential importance of the details of stochastic and non-stochastic dynamics, we constrain the parameter ranges explored by mutation to values taken from data on *Saccaromyces cerevisiae*. Different parameter values can cause the same network topology to display different dynamic behaviors; different topologies can also display similar dynamic behaviors^{21, 24-26}.”

METHODS

- I would think the work of Hermsen et al. 2006, Buchler et al. 2003, etc. are relevant to the methodology developed here. I leave it to the authors to decide if they want to cite these.

We thank the reviewer for drawing our attention to these two papers. We clearly see the similarity between our model and theirs. We agree with their conclusion that the distribution of TF binding sites can evolve to create different regulatory logics. We would love to cite these research, but the manuscript has reached the maximum number of references (70) allowed by *Nature Communications*, and indeed required us to remove quite a few between the past two submissions in order to reach that limit.

- p16, Evolutionary simulation. I disagree with the claim that modelling a heterogeneous population is out of the question. The new Fig 1 made me realise that every mutant is evaluated 200 times and every accepted mutant another 800 times. In reality, no-one is re-evaluated 200 times to establish a mean behavior. An organism is a single instance of a stochastic process. Thus a population of 200 individuals is computationally feasible, assuming that -- roughly speaking -- any tricks to optimize the re-evaluation of individuals cancel out against the 800 extra evaluations of an accepted mutant.

The key is to accept that not only evolution, but also development is a stochastic process. Surely individuals will get lucky some times and have a high fitness, while on average they would perform badly, but at some point their offspring will be unlucky and weeded out by selection.

The easiest solution is to remove the claim.

We have moved the claim to later in the Methods, now expanded and justified:

“Evolutionary simulations would require much more computation if we used a classic Wright-Fisher or Moran individual-based model, e.g. of population size 1000. Our scheme ensures that all mutations are evaluated by at least 200 gene expression simulations, making the probability of fixation of a beneficial mutation much higher than the $O(s)$ in an individual-based model. An

individual-based model would also require more than the 1,000 gene expression simulations required by our scheme per successful selective sweep.”

RESULTS

- I would invite the authors to consider the point of view that there may be a so-called complexity ratchet at work in their simulations. See Liard et al. 2018 for an example (https://www.mitpressjournals.org/doi/abs/10.1162/isal_a_00051).

It would mean that the networks that have an AND-gate C1-FFL do not evolve to/from other relatively high-fitness network structures. In some sense, one would have to "get lucky" at the start of evolution and hit upon an AND-gate FFL. When a network is not lucky, it wanders into an area of the fitness landscape that favors the use of weak TFBSs and alternative logic. Somehow that complexity of many weak binding sites obstructs the process of finding AND-gates.

If (and this is a big if) the above is the case, the claim that the preference for AND gates is associated to adaptation (as is done on page 22, l430), is still true, but not in the sense of adaptation vs mutation bias. Instead, adaptation can go into two directions, one simple and high-fitness, another complex and only sometimes high-fitness (i.e. signal controlled cases).

With respect to the three control conditions, the change in fitness function alters the shape of the landscape in terms of peaks and neutral/genotype networks (but not in terms of connectivity between individual genotypes). At first thought, the result that AND-gated FFLs are for filtering short signals, is not against the complexity ratchet. It is not specifically in favor either, so how these results fit together is a rather open question for me.

If it is easy to check what FFL logic precedes or comes after AND-gated FFLs over evolutionary time, I would be curious to know if that fits with the complexity ratchet idea. For the rest, I realize this is mostly me sharing a scientific discussion with the authors and other reviewer. There is no need for the authors to act upon this.

We aren't convinced whether a complexity ratchet is the best metaphor, vs. simply two adaptive peaks with one higher, but we're also not sure this matters for this conversation. In this paper, we aren't keen to get into the thorny issue of defining complexity.

In our simulations, both low-fitness replicates and high-fitness replicates start with a significant frequency of AND-gated C1-FFLs. These are quickly lost in the low-fitness replicates, but increase in frequency in the high-fitness ones. We think the increase is a combination of duplication and converting OR-gated C1-FFLs and slow-TF-controlled C1-FFLs. De novo evolution of AND-gated C1-FFLs may be rare, meaning that the low-fitness replicates don't recover from their early loss. Low-fitness replicates do have increasing frequencies of signal-controlled C1-FFLs, but these seem to be harder to convert to AND-gated. We have not added these results to the already-long paper.

- p29, l551. What mutational biases are meant?

Simply the ones given at the other end of the same sentence. We have rewritten to make our reasoning clearer.

- As remarked by the authors in their response, the manuscript is getting long. Unfortunately, I

agree as I find myself struggling through the second part of the results. The issue is not that it is boring, rather I lose track of the main message of the paper and the text is a rather technical report of subtle differences between each of the network motifs and their occurrence. If the authors see a way of focussing their message and leaving out unnecessary text, that would benefit their paper.

We feel the same problem! We have moved some details into the new Supplementary Notes 1 and 2, in order to shorten the main text and improve its flow.

MINOR CONCERNS

- **Title: would it not be more elegant to say "Under intrinsic noise, feed-forward regulation adaptively evolves via dynamics rather than topology"?**

Nature Communications does not allow punctuation in the title.

- **L55,74,478,479 etc. I do not see the need for quotes (").**

We have removed quotes from "AND" logic, "weak" binding sites, "ON"/"OFF" signals.

- **L98-100 Somehow the sentence does not read nicely, it has perhaps too much info. I would split it.**

We have split the sentence in two:

"We simulate the dynamics of TRNs as the TFs activate and repress one another's transcription over a timescale we refer to as "gene expression time". This generates the gene expression phenotypes on which selection acts over longer evolutionary timescales."

- **L106-111 This sentence is loooong.**

We have rewritten the sentence and the one before it:

"TF binding to the cis-regulatory sequence of a gene affects chromatin, which affects transcription rates, eventually feeding back to affect the concentration of TFs and hence their binding. Gene expression is further controlled by 5 gene-specific parameters: mean duration of transcriptional bursts, mRNA degradation rate, protein production rate, protein degradation rate, and gene length (which affects delays in transcription and translation). We model mutation to these 5 parameters, the cis-regulatory sequences, the consensus binding sequences, and the maximum binding affinity of TFs."

- **Fig 5A, 7A. Please add the legend of Fig 2 to remind me what the arrows mean.**

We have added a legend to Fig. 5a, 7a, 9a, Fig. S4, and Fig. S7a to explain the arrows.

- **l476. I find remembering what each of the three negative controls mean difficult. Would it be possible to add little cartoon explanations to Fig 6 that visualize how the fitness function is changed?**

We added Figure 6a to illustrate all selection conditions, except for the straightforward "no selection".

- **Fig 9. Is there a journal policy on sub-subpanels naming? Personally, I prefer to just name them sequentially, instead of hierarchically.**

Fig. 9 has been simplified, and we have removed the non-conventional labels on sub-subpanels.

SUPP INFO

- **Table S3. The heading of column 3-4 is a bit double, not? An accepted mutation is given to be accepted.**

Table S3 is now S6. We have changed the title to
"Probability that an accepted mutation is of this type"

- **Table S3, l36. typo in "which can be easily occur"**

Fixed

- **Table S3. Just for scientific discussion: mutations of locus length become a lot more popular in late evolution, even if they are a magnitude less frequent than gene indels. I am curious how the authors would explain that. I guess it is one of the few relatively neutral mutations on a gene's upstream region.**

The probability that a mutation to locus length is accepted, given it occurs, does not change much during evolution. The reason that there is an increase in the probability that an accepted mutation is to locus length is because the rate of locus length mutations is a function of locus length, and locus length increases during evolution. Specifically, the average locus length in the first 1,000 evolutionary steps is 393 codons, increasing to 501 codons in the last 1,000 steps.

- **Table S4. It would aid the reader if the authors explain why selection and no-selection in the ratio calculations of V and M are swapped? V_n/V_s , but M_s/M_n . I imagine because in the selection case, average parameter values tend to be greater than in the no-selection case. Variation, however, is larger in the no-selection case.**

Table S4 is now S3. We have elaborated on this in the table legend. The former gives an intuition for constraint, where high numbers indicate high constraint, while the latter gives an intuition as to whether selection favors large values vs. small.

- **Table S4, l46-51. The explanation of sampling could be better formulated.**

We split an overly long sentence into three, and rewrote for clarity.

- **Table S4, l61. Strictly speaking also for $R_{Act_to_Int}$, $V_n/V_s < 1.0$, not just for $K_d(0)$. Taking into account Table S5 and S6, I see that is an outlier.**

We have rewritten to reflect this.

- **Fig S12. Please check the naming of the motifs. I do not see a 'C' and 'A' is never mentioned. Also repeating the legend of Fig 2 would be a useful reminder.**

Fig. S12 is now Fig. S4. We have fixed the labels and repeated the legend.

- **Disclaimer: I have not had the time to carefully dig through all the supplementary methods.**

SOFTWARE

- I tried to compile the code with gcc, but it gets stuck at an icc specific flag, namely "-fp-model". If you claim that it should work with gcc, please fix this. Otherwise simply state that it is only guaranteed to work with icc.

We have revised the makefile. The icc specific flags are automatically removed when the compiler is gcc.

**With best wishes,
Anton Crombach**